# LEARNING TO RANK FOR AUTOML: ENHANCING PIPELINE SELECTION WITH RANKING INFORMATION

## ABSTRACT

This paper introduces a learning-to-rank (LTR) framework to address the problem of pipeline selection in automated machine learning systems. The traditional approach to AutoML involves learning to predict the performance of various pipelines on a given task based on data acquired from previous tasks (i.e., meta-learning), which can be complex due to the need for different models for each task-specific metric. The proposed framework aims to select the best pipeline based on ranking rather than estimating a target metric, aligning more closely with the ultimate goal of the task (i.e., selecting pipeline candidates in order, from more to least promising). This approach enables more robust, metric-agnostic solutions that are easier to compare using ranking metrics like NDCG and MRR. The paper evaluates LTR strategies on public OpenML datasets, demonstrating a clear advantage for ranking-based methods. Additionally, the integration of LTR with Bayesian optimization and Monte Carlo tree search is explored, leading to improvements in the ranking metrics. Finally, the study found a strong correlation between ranking metrics (e.g., NDCG and MRR) and AutoML metrics, such as the task objective metric and the time to find the best solution, providing insights into how ranking-based methods could enhance AutoML systems.

## 1 INTRODUCTION

Machine learning (ML) has become an essential component of countless scientific and industrial endeavors, ranging from healthcare (Waring et al., 2020) to e-commerce (Micu et al., 2019). However, applying machine learning in the real world is far from straightforward, as it involves multiple steps such as data preparation (preprocessing), feature and model selection, hyperparameter tuning, performance evaluation, etc. (Barbudo et al., 2023; Vazquez, 2022). The complexity in the design of machine-learning solutions led to the development of automated Machine Learning (AutoML) techniques that seek to relieve ML practitioners from repetitive and time-consuming tasks that have a large impact on the final performance of ML systems (Karmaker et al., 2021).

While AutoML helped widen the adoption of machine learning solutions at scale by simplifying the process of finding optimal configurations, it faces significant challenges. One such challenge is efficiently leveraging past experiences to improve performance on new tasks. This problem led to the development of meta-learning algorithms and techniques (Vanschoren, 2019) to capture the knowledge gained from solving a variety of tasks to help find optimal solutions for novel (future) ones. Meta-learning algorithms rely on meta-models to predict the performance of a given configuration (pipeline) on a specific dataset. This prediction serves as an approximation of the actual performance that the model would demonstrate if it were trained with the specified data. The benefit of this approach is that it prevents from actually training the model on the data in question, saving time and costs. Note that predicting pipeline performance involves predicting specific metrics for the different types of ML tasks (classification, regression, etc.), often requiring different models for each task-specific metric being optimized (Karmaker et al., 2021).

We can recognize two broad strategies in the design of an AutoML system. One is to build an optimal machine learning solution from a pool of simpler components (Ren et al., 2021), and the other is to select it from a set of predefined options or configurations (Yang et al., 2020). When the problem is about choosing among given solutions, it can be viewed as a ranking or recommendation task, where the goal is to rank the set of possible configurations according to their effectiveness in solving

the task. However, in many cases, the efforts of AutoML systems are focused on predicting the performance of configurations rather than tackling the selection or decision-making problem directly. For instance, when Bayesian optimization is used (Hutter et al., 2011), the selection of configurations often involves choosing the next most promising configuration based on a surrogate model. This model is designed to capture the relationship between the configuration and the target metric, to inform the selection process about which configuration to evaluate next. We argue that the selection problem could be addressed more effectively by framing it as a decision-making problem, thereby facilitating a more direct approach to determining the most valuable solutions.

Previous works explored different approaches to navigate the complex combinatorial space of available options (models, preprocessing steps, hyperparameters, etc.) in AutoML. Drawing parallels to finance portfolio management, some methods (Fusi et al., 2018; Yang et al., 2019) optimize model and pipeline selection based on historical performance, akin to optimizing a collection of assets. Other methods (Feurer et al., 2022; Laadan et al., 2019; 2020), directly adopt ranking and meta-learning strategies, demonstrating improvements in the efficiency of the AutoML workflow through data-driven decision-making and the pre-ranking of machine learning pipelines. However, although these approaches emphasize the importance of classifying information to speed up the search process and improve the efficiency of the AutoML system, they do not consider the underlying ranking problem explicitly.

This paper addresses some of these challenges by introducing a simple strategy for incorporating learning to rank (LTR) (Liu et al., 2009) into the pipeline selection problem in AutoML. Here, the aim is to learn to select the best candidate rather than estimating the target metric, which we believe is closer to the ultimate goal of the pipeline selection task. This approach allows us to generate more robust solutions that are agnostic to the target metric and easier to compare. Through the use of ranking metrics such as NDCG and MRR, we explore the effectiveness of different approaches, both score- and ranking-based. We also explore the use of LTR in classical sequential optimization techniques, such as Bayesian optimization and Monte Carlo tree search. The approaches are compared using public datasets from OpenML showing clear improvements for the ranking-based approaches.

The contributions of the paper are:

- The introduction of learning to rank (LTR) into the AutoML pipeline selection problem, aiming to more accurately reflect the ultimate goal of selecting the best candidate pipeline. This approach produces solutions that are robust and agnostic to the target metric, thereby simplifying comparisons.

- Evaluation of the effectiveness of different AutoML approaches using ranking metrics such as NDCG and MRR. This includes exploring the relationship between the ranking metrics and typical AutoML metrics, such as the improvement in the objective metric and the time it takes to find the best solution.

- Demonstrated improvements of ranking-based approaches over score-based ones through empirical evidence gathered from experiments on public datasets from OpenML. This underscores the efficiency and improved decision-making provided by adopting a ranking perspective in the AutoML workflow.

The rest of the paper is structured as follows. Section 2 introduces the learning to rank framework applied to AutoML, outlining some key concepts. Section 3 details the experimental setup, including the design and objectives of our study. Section 4 discusses the results of our experiments, providing insights into the performance of the LTR framework in the context of AutoML. Section 5 revisits related work. Section 6 discusses the findings and limitations of the approach. Finally, Section 7 concludes the paper and suggests avenues for future research.

## 2  LEARNING TO RANK FOR AUTOMATED MACHINE LEARNING

Learning to rank is an important concept in the field of machine learning that involves training models able to predict the relative order of a set of items. This technique is commonly used in search, recommendation systems, and other cases where prioritizing items based on their relevance is critical (Liu et al., 2009; Xia, 2019; Karatzoglou et al., 2013). This can be exploited in the context of AutoML through the concept of meta-learning (Vanschoren, 2019).

To prevent AutoML systems from repeatedly tackling similar tasks starting from scratch each time, meta-learning methods leverage data acquired from previous experience (meta-learning data) to improve performance on novel tasks (represented with meta-features). In our case, this corresponds to optimizing the procedure by which we select optimal pipelines for a wide variety of problems. We propose to do so by using ranking information instead of task-specific performance scores. In this way, the process of pipeline selection becomes independent of the absolute scores observed in the meta-learning data as well as from the different performance values achieved on each task.

In what follows, we first characterize the different approaches into score-based and ranking-based. While the former appears as the most common approach in AutoML, the latter provides a less explored and attractive alternative that will later prove to be effective.

## 2.1 SCORE-BASED MODELS

Score-based models represent most traditional approaches in the AutoML frameworks. These models aim to predict the target metric of a problem, such as classification accuracy or mean regression error. The predicted score is then used to select the most promising configuration or pipeline. This aligns with the pointwise approach in LTR theory, where the input space consists of individual configurations, and the output space is the score of these configurations on the target task.

This can be formalized as follows. Let $\mathcal{C} = \{C_1, C_2, \ldots, C_n\}$ represent the set of all possible configurations or pipelines in the AutoML search space. The goal is to predict the performance score of each configuration $C_i$ for a new given dataset $D$. To do so, we rely on a (parametric) score prediction model $f : \mathcal{C} \times \mathcal{D} \to \mathbb{R}$ that takes a candidate configuration $C_i$ and dataset $D$ as inputs and predicts a score $f(C_i, D)$ that relates to the predictive performance of $C_i$ on $D$. The score predicted by $f$ is specific to each task (accuracy, F1-score, regression errors, etc.). We assume we know the true value of this score for each pipeline-dataset combination in the meta-training dataset. We denote the ground-truth scores as $s(C_i, D)$. Training the meta-model is based on regression towards this metric by optimizing a suitable loss, such as the mean squared error of the scores, using a training dataset (e.g. mean squeare error of the scores) consisting of triplets of the form $\{(C_i, D_j, s_{ij})\}$, with $C_i \in \mathcal{C}$, $D_j \in \mathcal{D}$, and $s_{ij} = s(C_i, D_j)$.

## 2.2 RANKING-BASED MODELS

Ranking-based models focus on directly learning the ranking order of different configurations. These models are trained using explicit ranking information. The two primary formulations in this category are the pairwise and listwise approaches.

In the pairwise approach, the input space consists of pairs of configurations, $\mathcal{P} = \{(C_i, C_j) | C_i, C_j \in \mathcal{C}\}$, where $\mathcal{C}$ represents the set of all available pipeline configurations. In this case, we rely on a preference function $g : \mathcal{P} \times \mathcal{D} \to \{0, 1\}$ that given a pair of candidate configurations $(C_i, C_j)$ and a dataset $D$, it predicts which one is better: $g((C_i, C_j), D) = +1$ if $C_i$ is better than $C_j$, and 0 otherwise. The formulation could be extended to include a third category in which both configurations are deemed equally good. The binary case, however, has the advantage that it can be easily tackled as a binary classification problem. The model is trained using configuration pairs along with their relative ground-truth performances. For a new given dataset $D'$, configurations are ranked by comparing pairs and aggregating these pairwise preferences to form a global ranking. Common loss functions for the pairwise approach include hinge loss (Cao et al., 2006) or logistic loss (Burges et al., 2005), which explicitly account for a correct ordering of the pairs.

In the listwise approach, on the other hand, the input space corresponds to permutations of the set of all configurations $\mathcal{C}$. Instead of directly working with the set of all permutations of $\mathcal{C}$, we can think of a ranking function $h : \mathcal{C} \times \mathcal{D} \to \mathbb{R}^{|\mathcal{C}|}$ that takes as input the set of available configurations and a dataset $D$, and outputs a vector of scores of dimensionality $|\mathcal{C}|$, whose $i$-th entry corresponds to the relevance of configuration $C_i$ for solving the problem. Sorting the elements of this vector provides the permutation indices that rank the elements of $\mathcal{C}$ according to their relevance for $D$. Training such a model involves using the entire set of configurations $\mathcal{C}$ and their rankings for different datasets. The model learns to predict these rankings as accurately as possible. There are two types of output spaces used in the listwise approach. In the first, for a new dataset $D'$, the model predicts a ranking position for each configuration. The configurations are then ranked based on these positions (Taylor et al.,

2008). In the second, the output space contains the ranked list (or permutation) of the configurations (Cao et al., 2007). In this work, we will focus on the first one. By transforming the input and output spaces from a score into ranking positions, we have a simple way to compare pipeline selection approaches with and without ranking information. Finally, different losses can be used to train a ranking model listwise (Cao et al., 2007; Yue et al., 2007; Taylor et al., 2008), all of which extend the ranking formalism to account for a large set of candidate objects.

### 2.3 INTEGRATING LTR INTO SEQUENTIAL OPTIMIZATION FRAMEWORKS

LTR approaches described so far require evaluating the whole set of configurations during inference. With large configuration spaces, this might result in excessive computations and time. Sequential model-based optimization (SMBO) stands as an alternative formulation that seeks to improve the efficiency of searching for optimal configurations in complex spaces. SMBO approaches rely on discovering new configurations by using an estimate of the target metric as a comparison reference. In the Bayesian optimization (BO) approach proposed by Hutter et al. (2011), a surrogate model is trained to estimate the value of the target metric for a given configuration. In this way, configuration pipelines are sampled through the use of an acquisition function, which guides the search for the optimal solution by quantifying the expected utility of selecting a particular configuration. The configuration that exhibits the highest estimated value according to this function is then selected.

In our work, we consider the selection problem as a ranking process that we can directly integrate into the model. For instance, if our surrogate model is a regression of the accuracy for a classification task, we can replace it with a regression that estimates the ranking position of the configuration. It is also possible to use a pairwise model that, from a sample of configurations, orders them by comparing pairs and selects the best one. By replacing the surrogate model with a pre-trained ranking model we expect to improve overall performance. We call this variant of the BO algorithm BO-Rank.

Similarly, we can also reformulate model-free approaches such as Monte Carlo Tree Search (MCTS) Vazquez et al. (2022). In MCTS, the search space is encoded as a tree whose leaves encode the set of possible configurations. Configurations are compared using a selection policy, which is computed based on the target metric for the task, often referred to as the reward. In this case, we can replace the objective function with a ranking objective, i.e. by using the ranking position as the reward from which the selection policy is computed. This allows us to precompute the partial ranks for each configuration and thus avoid the need to compute estimates of the target metric on a validation set as in the traditional MCTS. We call this variant of the algorithm MCTS-Rank.

Finally, we also note that our approach could also be applied in the case of model-based MCTS by using a ranking model as a surrogate during the simulation step. However, for the sake of simplicity, we leave this to be explored in future works.

## 3 EXPERIMENTAL SETUP

Our evaluation encompassed a comprehensive comparison of four categories of selection approaches:

1. **Random Selection:** A baseline approach involving the random selection of pipelines, serving as a control to gauge the effectiveness of more sophisticated methods.

2. **Average Best:** This approach involves selecting the best pipeline by ranking them based on their performance in the training data.

3. **Regression:** This category uses regression to predict the performance metric, treating the problem as the prediction of a score Hutter et al. (2011); Feurer et al. (2015), or to predict the rank of pipelines in a list, treating the problem as the prediction of a position.

4. **Sequential Optimization:** We consider the problem of Sequential Model-Based Optimization (SMBO) using classical optimization frameworks, Bayesian Optimization (BO) and Monte Carlo Tree Search (MCTS).

Each category was analyzed in two flavors: score-based and ranking-based (except for the Random Selection baseline). In the case of Average Best (Avg), one version creates a list of the best pipelines ordered by average score, while the other orders them by average rank. For the regression-based

approaches, regressors from scikit-learn and LightGBM (LGBM) were used with default hyperparameters to avoid bias from hyperparameter tuning. The regressors utilized were LinearRegression (LR), Lasso, Ridge, RandomForestRegressor (RF), GradientBoostingRegressor (GB), and LGBMRegressor (LGBM).

Additionally, for the SMBO frameworks, BO and MCTS were employed. These frameworks were selected for their efficacy in navigating large search spaces, providing a robust platform to evaluate the comparative performance of the proposed ranking methodologies. This experiment allows us to evaluate and compare the effectiveness of different approaches in the context of continuous pipeline selection within traditional frameworks, offering insights into the potential of these methods to enhance and optimize existing techniques.

For BO, we consider a straightforward setting in which we initialize the process with a pre-trained (meta-learned) surrogate model. Specifically, we first train a linear regression model on training tasks, where it learns to either estimate the score of the pipelines (score-based regression) or their ranking position (ranking-based regression). In each iteration, we take a pipelines sample (10 for AMLB and 100 for OpenML-Weka) and employ a greedy approach to select the best next pipeline to evaluate as the acquisition function. After evaluating a pipeline, the surrogate model is retrained with the new information, and the pipeline is removed from the pool of possible options. The algorithm runs until there are no more pipelines available. The ordered list of selected pipelines is considered the final ranking of pipelines to evaluate.

Similarly, for MCTS, we use the model-free variant; that is, no surrogate model is used. Instead, we consider, in one case, a pre-calculated (meta-learned) average score of the pipelines as the reward function, and in the other case, the average ranking position. In each iteration of MCTS, we use a greedy approach for the rollout policy and the Upper Confidence Bound (UCB) for the selection policy (with $C = \frac{1}{\sqrt{2}}$) to discover the next pipeline to evaluate. After evaluating a pipeline, it is pruned from the pool of possible options. The algorithm runs until there are no more pipelines available. The ordered list of selected pipelines is considered the ranking of pipelines to evaluate.

We conducted experiments on three different scenarios originating from a set of datasets extracted from the OpenML initiative (Vanschoren et al., 2014). For our experiments, a scenario is formed by: a set of tasks, a set of features that describe the tasks, a set of models evaluated on those tasks, and finally, an objective metric. Table 1 describes the scenarios. The first one was taken from the ASLib initiative Bischl et al. (2016), specifically, the OPENML-WEKA-2017 scenario. This set consists of 105 classification tasks solved by 30 models. OPENML-WEKA-2017 includes features describing datasets and the predictive accuracy reported for evaluating models on tasks.

| N° | Dataset | #Tasks | #Models | Objective Metric |
|---|---|---|---|---|
| 1 | ASLib - OPENML-WEKA-2017 | 105 | 30 | Predictive Accuracy |
| 2 | AMLB Classification 2023 | 71 | 2160 | Balanced Accuracy |
| 3 | AMLB Regression 2023 | 33 | 1485 | Negative RMSE |

Table 1: Summary of the scenarios evaluated.

The second and third scenarios consist of tasks from the Automated Machine Learning Benchmark (AMLB) (Gijsbers et al., 2022), an open and extensible benchmark that follows best practices. This set of OpenML tasks includes 104 tasks, split between 71 classification and 33 regression problems, ensuring a broad representation of common challenges in machine learning. The complete list of OpenML task IDs can be found in Appendix A. With these tasks, we created two different scenarios consisting of either classification or regression tasks.

To collect pipeline performances for the AMLB scenarios, we executed 2160 different pipelines for classification and 1485 for regression. For the purpose of this experiment, we executed all pipelines in a search space defined by a simple grammar (Appendix C) as in Vazquez et al. (2022). The execution was carried out on an Amazon EC2 R5 spot instance (8 vCPUs and 64GB of RAM), and results were saved, with the sole restriction that each pipeline must be completed within one minute. Each pipeline was then identified by a randomly assigned ID. We used standard train-test splits, as provided by each OpenML task. For classification and regression problems, balanced accuracy and root mean square error (RMSE) were used as metrics, respectively.

To describe the datasets, we computed the set of meta-features proposed by Rivolli et al. (2022) with the Python Meta-Feature Extractor (Alcobaça et al., 2020)[1]. These meta-features encode different characteristics useful for representing machine learning datasets. We applied PCA to reduce the dimensionality of such representations to three dimensions. We found that this value preserves most of the discriminative power of the original representation while considerably reducing the number of dimensions. Additionally, we considered an additional meta-feature to capture the average performance of each configuration. This additional meta-feature corresponds to the average performance score for score-based models and the average rank for ranking-based ones. The average is computed on the training dataset.

For learning to rank (i.e., meta-learning), we used task-level cross-validation with 10 folds. Additionally, we ran experiments with 10 different seeds. For evaluation, we considered two classical ranking metrics: Normalized Discounted Cumulative Gain (NDCG) and Mean Reciprocal Rank (MRR). NDCG measures the quality of the ranking system in terms of the position of the top-K elements in a ranked list, with a diminishing value for elements further down the list. In contrast, MRR considers the position of the highest-ranked pipeline, computed as the inverse of the position of the best configuration.

In addition, we measured important AutoML system metrics across the multiple tasks: the average score (SCORE) representing the mean value of the objective metric (such as accuracy or RMSE); the time to find the best solution (TTB) measuring how long it takes for the system to identify the best-performing pipeline within the predefined search space; and the average rank (AVG RANK) reflecting the ranking of each approach performance relative to others (score- vs ranking-based) where a lower rank indicates better performance (e.g., the best-performing framework on a task is assigned a rank of 1, the second-best a rank of 2, etc.).

# 4 RESULTS

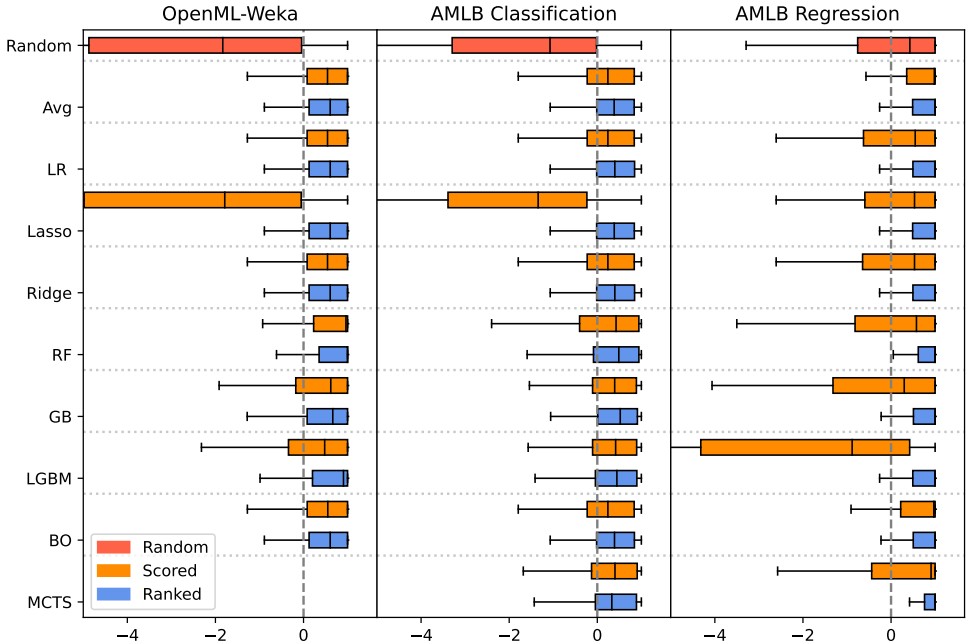

Figure 1: Boxplots of variants (Rank-based in blue and Score-based in orange) performance (x-axis) across tasks for each predictor (y-axis) after scaling the performance values from the mean (0) to best value observed.

---

[1] https://pymfe.readthedocs.io/

Figure 1 shows the performance distribution for the different system variants (Rank- and Score-based) across multiple tasks, with the performance values scaled from the mean to the best value observed. The predictors include various machine learning models such as MCTS, BO, LGBM, GB, RF, Ridge, Lasso, and LR, as well as the Avg best strategy and baseline Random selection on different datasets or scenarios (e.g., OpenML-Weka, AMLB Classification, AMLB Regression).

The inclusion of the random selection baselines shows that although both score- and ranking-based variants perform generally better, overall performance depends to a greater extent on the type of predictor. For most predictors, the Rank-based variants (blue) tend to exhibit a more consistent and narrower distribution compared to the Score-based variants (orange), indicating more reliable performance across tasks. This suggests that the Rank-based variant might be more robust when applied to different tasks. In addition to its lower variability, the Rank-based variant can achieve higher performances. This difference is particularly noticeable in the AMLB Regression dataset, while in the other datasets, the difference is more subtle.

Specific predictors, such as Light Gradient Boosting Machine (LGBM) and Random Forest (RF), exhibit tighter and more favorable performance distributions for both variants, suggesting these models are robust across different tasks. However, simpler models like LR and Avg show narrower performance ranges in Score-based variants. This advantage tends to disappear when the target is transformed to a Rank-based variant. This could indicate that predicting scores is more susceptible to overfitting, while predicting ranking positions is more robust.This is clearly visible in the AMLB Regression, although it can also be seen in OpenML-Weka.

Table 2: Comparison of different approaches evaluated across three datasets: OpenML-Weka (D1), AMLB Classification (D2), and AMLB Regression (D3). Each metric is assessed at three positions ("a", "b", and "c"), measuring performance in terms of NDCG, MRR, SCORE, RANK, and TTB. For the AMLB datasets, the positions are 1, 10, and 100, while for the OpenML-Weka dataset, the positions are 1, 5, and 10. Cell color indicates where the Rank-based approach improves over the Score-based approach (green ■ > 0, dark green ■ > 10%, and blue ■ > 50% of improvements), decreases (yellow ■ < 0, orange ■ < 10%, and red ■ < 50% of improvements), or shows no changes (grey ■). The symbols "‹" and "«" indicate statistical significance and "–" indicates not evaluated.

| | | NDCG@ | | | MRR@ | | | SCORE@ | | | RANK@ | | | TBB@ | | |
|---|---|---|---|---|---|---|---|---|---|---|---|---|---|---|---|---|
| | | a | b | c | a | b | c | a | b | c | a | b | c | a | b | c |
| D1 | Avg | « | « | « | « | « | « | « | « | « | « | « | « | – | – | – |
| | LR | « | « | « | « | « | « | « | « | « | « | « | « | – | – | – |
| | Lasso | « | « | « | « | « | « | « | « | « | « | « | « | – | – | – |
| | Ridge | « | « | « | « | « | « | « | « | « | « | « | « | – | – | – |
| | RF | « | « | « | « | « | « | « | « | « | « | « | « | – | – | – |
| | GB | « | « | « | | | « | « | « | « | « | « | « | – | – | – |
| | LGBM | « | « | « | « | « | « | « | « | « | « | « | « | – | – | – |
| | BO | « | « | « | « | « | « | « | « | « | « | « | « | – | – | – |
| | MCTS | – | – | – | – | – | – | – | – | – | – | – | – | – | – | – |
| D2 | Avg | « | « | « | ‹ | ‹ | ‹ | | | | | | | | | |
| | LR | « | « | « | ‹ | ‹ | ‹ | | | | | | | | | |
| | Lasso | « | « | « | « | « | « | « | | « | « | « | « | « | « | « |
| | Ridge | « | « | « | ‹ | ‹ | ‹ | | | | | | | | | |
| | RF | « | « | « | | | | | | | | | | | | |
| | GB | « | « | « | | | | | | | | | | | | |
| | LGBM | « | « | « | | | | | | | | | | | | |
| | BO | « | « | « | « | « | « | | | | | | | | | |
| | MCTS | | | | | | | ‹ | ‹ | ‹ | | | | | | |
| D3 | Avg | « | « | « | « | « | « | « | « | « | ‹ | ‹ | ‹ | « | « | « |
| | LR | « | « | « | « | « | « | « | « | « | « | « | « | « | « | « |
| | Lasso | « | « | « | « | « | « | « | « | « | « | « | « | « | « | « |
| | Ridge | « | « | « | « | « | « | « | « | « | « | « | « | « | « | « |
| | RF | « | « | « | « | « | « | « | « | « | « | « | « | « | « | « |
| | GB | « | « | « | « | « | « | « | « | « | « | « | « | « | « | « |
| | LGBM | « | « | « | « | « | « | « | « | « | « | « | « | « | « | « |
| | BO | « | « | « | « | « | « | « | « | « | « | « | « | « | « | « |
| | MCTS | « | « | « | « | « | « | « | « | « | « | « | « | « | « | « |

Table 2 presents a detailed comparison of different approaches evaluated across the three datasets: OpenML-Weka (D1), AMLB Classification (D2), and AMLB Regression (D3). The performance of these approaches is measured using five metrics (i.e., NDCG, MRR, SCORE, RANK, and TTB) across three cutoff points specific to each dataset (1, 5, and 10 for OpenML-Weka, and 1, 10, and 100 for the AMLB datasets). The table also uses color coding to indicate where the Rank-based approach improves over the Score-based approach. The symbols "‹" and "«" indicate statistical significance using the Wilcoxon signed-rank test (p-value < 0.05) and after Bonferroni adjustment (p-value < 0.005), respectively. The missing entries ("-") for D1 are due to the dataset's lack of temporal information, which makes it impossible to calculate TTB, and its absence of information about the components that form the pipelines, preventing the execution of MCTS, as MCTS requires knowledge of pipeline components to structure exploration as a decision tree.

Across all metrics and datasets, the green/blue cells indicate areas where the Rank-based approach outperforms the Score-based approach. These improvements are consistently observed across multiple predictors, suggesting that the enhancements introduced are robust across different metrics and datasets. For NDCG at all positions, significant improvements are observed across most predictors.

In terms of the SCORE, MRR, and RANK metrics, significant improvements are also observed, particularly at lower positions (e.g., SCORE@a, MRR@a, and RANK@a). For TTB, however, the improvement appears to increase as the number of positions considered increases. This indicates that the evaluated methods are time-effective while maintaining performance even as the cutoff position increases.

Yellow cells indicate metrics where a decrease in performance was observed. Interestingly, these are relatively sparse, highlighting that the evaluated approaches generally lead to better performance across most metrics. This decrease is less than 10%; only MRR in RF on the AMLB Classification dataset shows a decrease in performance exceeding 10%, and none exceed 50%. In particular, in the AMLB Classification dataset, there are a majority of instances where metrics do not show improvement at certain positions, with SCORE@b and RANK@b being the most frequently observed. However, it should be noted that in some cases, there is statistical significance indicating that the paired results are generally better, even though the average performance is lower.

The presence of statistical significance symbols ("‹" and "«") across many of the metrics highlights that the observed improvements are not just by chance but are statistically significant. This is particularly evident in datasets D1 and D3, where the majority of improvements across different datasets and positions are statistically significant. The stronger significance (denoted by "«") after Bonferroni adjustment further emphasizes the robustness of the findings, indicating that the improvements are consistent and reliable across multiple tests (i.e., across different seeds). A summary of all experiments corrected using the Bonferroni method can be found in Appendix B.5.

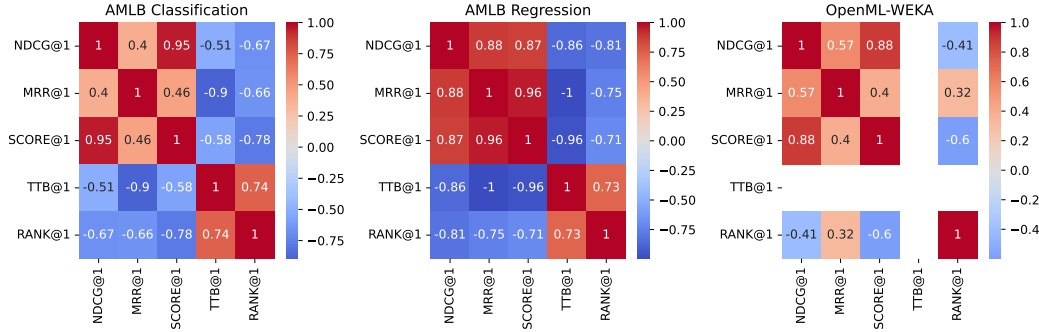

Figure 2: Spearman correlation between metrics at first position.

Figure 2 illustrates the correlation between different performance metrics at the first position across three datasets: AMLB Classification, AMLB Regression, and OpenML-WEKA. The metrics evaluated include NDCG@1, MRR@1, SCORE@1, TTB@1, and RANK@1. The correlation values, which range from -1 to 1, indicate the strength and direction of the linear relationships between these metrics.

Across all datasets, there is a strong positive correlation between NDCG@1 and SCORE@1. For instance, in the AMLB Classification dataset, the correlations between these metrics are particularly high, with values such as 0.95 between NDCG@1 and SCORE@1. This suggests that these metrics are closely aligned in assessing the quality of the top-ranked predictions. The implication is that improvements in one of these metrics are likely to be reflected in the others, indicating their consistency in evaluating the effectiveness of ranking at the top position.

Another interesting finding is that TTB@1 shows a strong negative correlation with MRR@1, particularly in the AMLB datasets where it was evaluated. For example, in AMLB Regression, the correlation between TTB@1 and MRR@1 is -1. This negative correlation indicates that when MRR@1 improves, TTB@1, which focuses more on the time to find the best solution, tends to decrease. This is expected since MRR tends to position the best result at the top, and optimizing it may lead to finding it faster in terms of time. RANK@1 exhibits a negative correlation with NDCG@1, MRR@1, and SCORE@1 across all datasets. In the AMLB Classification dataset, for instance, RANK@1 correlates negatively with NDCG@1 (-0.67) and MRR@1 (-0.66). This inverse relationship implies that as the average rank (RANK@1) improves (lower rank value), the other metrics show better performance (higher values), reinforcing the idea that these metrics are effective in capturing the quality of the ranking, particularly at the top positions.

The correlations exhibit some variation across different datasets. For example, in the OpenML-WEKA dataset, the correlation between MRR@1 and RANK@1 is 0.32, much weaker and with the opposite sign compared to the AMLB datasets. This variability could suggest that the relationships between these metrics may be affected by the length of the models evaluated (only 20 in OpenML-WEKA), but this is speculative.

In general, improvements in ranking-based strategies compared to those that rely on score-based models are clear. The underperformance of score-based models, although better than random selection, highlights the advantages of using ranking information. The results suggest that the choice between ranking-based and score-based variants can significantly influence the consistency and robustness of performance across tasks for optimal pipeline selection in AutoML problems.

## 5 RELATED WORK

Pipeline selection is an important problem of AutoML. Significant approaches have been developed to efficiently navigate the combinatorial space of possible models, preprocessing steps, and hyperparameters (Hutter et al., 2011; Feurer et al., 2015; Vanschoren, 2019). In particular, Vanschoren 2019 categorizes meta-models into performance predictors and ranking generators. We extend this idea by showing that transitioning from score-based to ranking-based methods in classical approaches like BO and MCTS could lead to improvements in pipeline selection.

Furthermore, some other approaches have evaluated the possibility of using ranking techniques to improve the selection of pipelines. For instance, AutoFolio (Lindauer et al., 2015) uses several algorithms based on performance scores to train a model based on portfolio training data, like pairwise classifiers. While Autofolio proposes some transformations to the target like log (Xu et al. (2008)) or z-score normalization (Collautti et al. (2013)) to improve the selection, none of these consider the ranking position. Similarly, other works (Sun & Pfahringer, 2013; Tornede et al., 2020) use comparisons as a way to learn better models for algorithm selection. However, comparisons are primarily score-based between performance scores and not between ranking positions.

An interesting aspect is that, in practice, the pipeline selection problem can occur before the system's search process. For example, AutoGluon (Erickson et al., 2020) employs an ensemble strategy to combine predictions from different types of preselected fixed pipelines. This methodology leverages the strengths of various models to improve overall performance, albeit within the limitations of the preselected pipeline space. On the other hand, GramML (Vazquez et al., 2022) adopts a different strategy by learning to delineate the space while exploring it using MCTS. It also extends the search approach to preselected hyperparameters within fixed and predefined ranges (Vázquez et al., 2023). Preselection allows a priori knowledge of the size of the space and the constituent pipelines, facilitating a more directed and potentially efficient search.

The idea of working with predefined pipeline spaces resemble to the concept of portfolio management in finance, where a collection of assets is optimized to achieve the best possible return for a given level

of risk. Thus, approaches like Probabilistic Matrix Factorization (Fusi et al., 2018) and Oboe (Yang et al., 2019), seek to optimize the selection of models and pipelines based on historical performance data. These systems leverage collaborative filtering techniques to predict the performance of various models on new datasets, thereby guiding the selection process in a data-driven manner. By building on the successes and failures of past model applications, portfolio-based approaches aim to streamline the AutoML workflow and enhance decision-making efficiency.

Another innovative direction in AutoML research involves using ranking information to improve the cold start problem and meta-learning strategies. Auto-sklearn 2.0 (Feurer et al., 2022), for example, uses portfolio information to prioritize models and configurations that are likely to perform well, based on historical performance on similar tasks. This approach helps to quickly identify promising starting points for the BO process, reducing the time and resources required for model selection. It also uses preselected pipelines to address the cold start problem.

Similarly, RankML (Laadan et al., 2019) explicitly focusing on a meta-learning approach for the pre-ranking of machine learning pipelines. This methodology aims to leverage accumulated knowledge to predict the efficacy of different pipelines before detailed evaluations, potentially saving significant computational resources and accelerating the AutoML process.

MetaTPOT (Laadan et al., 2020) takes this concept to enhances the evolutionary algorithm-based optimization tool (TPOT, Olson & Moore (2016)) by incorporating meta-learning techniques that use ranking information to guide the search process. By learning from the performance rankings of pipelines across a variety of datasets, MetaTPOT seeks to improve upon its predecessor by identifying efficient pipelines more quickly.

The approaches described above recognize the importance of using ranking information, whether to preselect the system's components, order them to expedite the search, or as a form of meta-learning to enhance system efficiency. However, the problem of learning to rank and the importance of using positional information has not been addressed, to our knowledge, until this work. We believe this work presents a novel conceptual framework as a basis for learning to rank in AutoML that can potentially be transferred to many other approaches.

## 6 Conclusions and Discussion

This work presents a simple strategy to frame the pipeline selection task in automated machine learning (AutoML) as a Learning-to-Rank (LTR) problem. Based on the hypothesis that incorporating the positional information of pipelines from previous results can improve selection outcomes, this work compares traditional pipeline selection approaches (score-based) against a ranking-based counterpart by transforming the prediction objective from a target metric to a ranking position.

The results of the experiments demonstrate a marked improvement of the ranking-based approach over the score-based approach. The superior performance underscores the importance of considering the positional information rather than focusing on isolated metric predictions regarding to the ranking problem. Furthermore, it was found that metrics commonly used in ranking problems correlate with metrics more typical of AutoML, such as the objective metric and the time taken to find the best solution. This finding corroborates the hypothesis that ranking optimization strategies are more aligned with the intrinsic structure of selection problems in AutoML.

However, this study primarily focuses on demonstrating the effectiveness of ranking-based approaches in scenarios where all pipelines are known, which means that the search space is finite and predefined. While we believe this scenario addresses a significant portion of the pipeline selection problem, as predefined pipelines are common in many practical tools, further studies are needed to extend these findings to more diverse scenarios, including feature selection and hyperparameter optimization. Additionally, evaluating more complex and computationally expensive LTR algorithms, such as pairwise methods, was beyond the scope of this work.

Lastly, while the tasks and pre-computed pipelines used for evaluation may not capture the full range of possibilities in the AutoML landscape, we believe that by leveraging classification and regression tasks from OpenML and established benchmarks like AMLB and ASLib, the results are generalizable and can be built upon in future research.

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

## A  OPENML TASKS

This appendix provides additional details on the specific datasets used in our study. OpenML tasks are separated by scenario into OpenML-Weka tasks, AMLB-Classification tasks and AMLB-Regression tasks.

### A.1  OPENML-WEKA TASKS

The OpenML task IDs for OpenML-Weka 2017 as described in ASLib initiative are:

2097, 2098, 2102, 1701, 1702, 1705, 1710, 1711, 1713, 1714, 1715, 1717, 1719, 1720, 1721, 1722, 1723, 1727, 1728, 1730, 1731, 1735, 1736, 1740, 1742, 1744, 1752, 1757, 1764, 10041, 10045, 10046, 10047, 10050, 10053, 10054, 10055, 10067, 10069, 10070, 10071, 10072, 10074, 10075, 10076, 10077, 10079, 10080, 10082, 10083, 10084, 10085, 10086, 7532, 7535, 7536, 125849, 125850, 125851, 125852, 125853, 125854, 125855, 125857, 125859, 125861, 125865, 125866, 125867, 125868, 125869, 125870, 125871, 125873, 125874, 125875, 125876, 125877, 125878, 125879, 125880, 125881, 125884, 125885, 125886, 125887, 125888, 125889, 125890, 125891, 125892, 125894, 125897, 125898, 125899, 125901, 125902, 125905, 125906, 125909, 125910, 125911, 125913, 125914, 125915

We excluded from the selectable datasets those from the AutoML Benchmark (AMLB) that also met the criteria, as they are used in the other experiment. The excluded dataset IDs are as follows: 146818, 359955, and 190146.

### A.2  AMLB-CLASSIFICATION TASKS

The selected AMLB IDs for classification tasks are:

2073, 3945, 7593, 10090, 146818, 146820, 167120, 168350, 168757, 168784, 168868, 168909, 168910, 168911, 189354, 189355, 189356, 189922, 190137, 190146, 190392, 190410, 190411, 190412, 211979, 211986, 359953, 359954, 359955, 359956, 359957, 359958, 359959, 359960, 359961, 359962, 359963, 359964, 359965, 359966, 359967, 359968, 359969, 359970, 359971, 359972, 359973, 359974, 359975, 359976, 359977, 359979, 359980, 359981, 359982, 359983, 359984, 359985, 359986, 359987, 359988, 359989, 359990, 359991, 359992, 359993, 359994, 360112, 360113, 360114, 360975

### A.3  AMLB-REGRESSION TASKS

The selected AMLB IDs for regression tasks are:

167210, 233211, 233212, 233213, 233214, 233215, 317614, 359929, 359930, 359932, 359933, 359934, 359935, 359936, 359937, 359938, 359939, 359940, 359941, 359942, 359943, 359944, 359945, 359946, 359948, 359949, 359950, 359951, 359952, 360932, 360933, 360945

Task 359931 was excluded due to errors.

## B DETAILED EXPERIMENTS RESULTS

In this section we break down the results presented in Section 4, separating them into scenarios OpenML-Weka, AMLB-Classification and AMLB-Regression problems.

### B.1 OPENML-WEKA

Table 3: Comparison of approaches using ranking metrics NDCG and MRR on ASLib tasks. The best results in each group are highlighted in bold.

| | var | NDCG@1 | NDCG@5 | NDCG@10 | MRR@1 | MRR@5 | MRR@10 |
|---|---|---|---|---|---|---|---|
| Random | | $0.600^{(0.27)}$ | $0.638^{(0.16)}$ | $0.671^{(0.13)}$ | $0.068^{(0.25)}$ | $0.123^{(0.27)}$ | $0.146^{(0.26)}$ |
| Avg | Score | $0.872^{(0.10)}$ | $0.863^{(0.08)}$ | $0.858^{(0.07)}$ | $0.128^{(0.33)}$ | $0.301^{(0.34)}$ | $0.318^{(0.32)}$ |
| | Rank | $0.879^{(0.09)}$ | $0.865^{(0.08)}$ | $0.860^{(0.06)}$ | $0.133^{(0.34)}$ | $0.304^{(0.34)}$ | $0.330^{(0.32)}$ |
| LR | Score | $0.872^{(0.10)}$ | $0.863^{(0.08)}$ | $0.858^{(0.07)}$ | $0.128^{(0.33)}$ | $0.301^{(0.34)}$ | $0.318^{(0.32)}$ |
| | Rank | $0.879^{(0.09)}$ | $0.865^{(0.08)}$ | $0.860^{(0.06)}$ | $0.133^{(0.34)}$ | $0.304^{(0.34)}$ | $0.330^{(0.32)}$ |
| Lasso | Score | $0.610^{(0.26)}$ | $0.638^{(0.16)}$ | $0.668^{(0.13)}$ | $0.076^{(0.27)}$ | $0.144^{(0.29)}$ | $0.165^{(0.28)}$ |
| | Rank | $0.879^{(0.09)}$ | $0.865^{(0.08)}$ | $0.860^{(0.06)}$ | $0.133^{(0.34)}$ | $0.304^{(0.34)}$ | $0.330^{(0.32)}$ |
| Ridge | Score | $0.872^{(0.10)}$ | $0.863^{(0.08)}$ | $0.858^{(0.07)}$ | $0.128^{(0.33)}$ | $0.301^{(0.34)}$ | $0.318^{(0.32)}$ |
| | Rank | $0.879^{(0.09)}$ | $0.865^{(0.08)}$ | $0.860^{(0.06)}$ | $0.133^{(0.34)}$ | $0.304^{(0.34)}$ | $0.330^{(0.32)}$ |
| RF | Score | $0.884^{(0.14)}$ | $0.878^{(0.09)}$ | $0.881^{(0.07)}$ | $0.253^{(0.43)}$ | $0.383^{(0.40)}$ | $0.403^{(0.38)}$ |
| | Rank | $0.899^{(0.13)}$ | $0.892^{(0.08)}$ | $0.898^{(0.06)}$ | $0.296^{(0.46)}$ | $0.426^{(0.41)}$ | $0.448^{(0.39)}$ |
| GB | Score | $0.846^{(0.16)}$ | $0.853^{(0.10)}$ | $0.854^{(0.07)}$ | $0.201^{(0.40)}$ | $0.333^{(0.38)}$ | $0.352^{(0.37)}$ |
| | Rank | $0.872^{(0.12)}$ | $0.867^{(0.08)}$ | $0.866^{(0.06)}$ | $0.156^{(0.36)}$ | $0.321^{(0.35)}$ | $0.344^{(0.34)}$ |
| LGBM | Score | $0.830^{(0.17)}$ | $0.850^{(0.10)}$ | $0.865^{(0.08)}$ | $0.177^{(0.38)}$ | $0.306^{(0.37)}$ | $0.333^{(0.35)}$ |
| | Rank | $0.889^{(0.12)}$ | $0.888^{(0.07)}$ | $0.894^{(0.06)}$ | $0.221^{(0.41)}$ | $0.383^{(0.38)}$ | $0.402^{(0.36)}$ |
| BO | Score | $0.872^{(0.10)}$ | $0.863^{(0.08)}$ | $0.858^{(0.07)}$ | $0.128^{(0.33)}$ | $0.301^{(0.34)}$ | $0.318^{(0.32)}$ |
| | Rank | $0.879^{(0.09)}$ | $0.865^{(0.08)}$ | $0.860^{(0.06)}$ | $0.133^{(0.34)}$ | $0.304^{(0.34)}$ | $0.330^{(0.32)}$ |

Table 4: Comparison of approaches using AutoML metrics average SCORE and average RANK on ASLib tasks. The best results in each group are highlighted in bold.

| | var | SCORE@1 | SCORE@5 | SCORE@10 | RANK@1 | RANK@5 | RANK@10 |
|---|---|---|---|---|---|---|---|
| Random | | $0.765^{(0.23)}$ | $0.856^{(0.17)}$ | $0.866^{(0.17)}$ | - | - | - |
| Avg | Score | $0.854^{(0.18)}$ | $0.869^{(0.17)}$ | $0.873^{(0.17)}$ | $1.041^{(0.20)}$ | $1.006^{(0.08)}$ | $1.107^{(0.31)}$ |
| | Rank | $0.856^{(0.18)}$ | $0.869^{(0.17)}$ | $0.874^{(0.17)}$ | $1.014^{(0.12)}$ | $1.002^{(0.04)}$ | $1.025^{(0.16)}$ |
| LR | Score | $0.854^{(0.18)}$ | $0.869^{(0.17)}$ | $0.873^{(0.17)}$ | $1.041^{(0.20)}$ | $1.006^{(0.08)}$ | $1.109^{(0.31)}$ |
| | Rank | $0.856^{(0.18)}$ | $0.869^{(0.17)}$ | $0.874^{(0.17)}$ | $1.014^{(0.12)}$ | $1.002^{(0.04)}$ | $1.027^{(0.16)}$ |
| Lasso | Score | $0.767^{(0.24)}$ | $0.854^{(0.18)}$ | $0.865^{(0.17)}$ | $1.784^{(0.41)}$ | $1.632^{(0.48)}$ | $1.524^{(0.50)}$ |
| | Rank | $0.856^{(0.18)}$ | $0.869^{(0.17)}$ | $0.874^{(0.17)}$ | $1.146^{(0.35)}$ | $1.137^{(0.34)}$ | $1.103^{(0.30)}$ |
| Ridge | Score | $0.854^{(0.18)}$ | $0.869^{(0.17)}$ | $0.873^{(0.17)}$ | $1.041^{(0.20)}$ | $1.006^{(0.08)}$ | $1.109^{(0.31)}$ |
| | Rank | $0.856^{(0.18)}$ | $0.869^{(0.17)}$ | $0.874^{(0.17)}$ | $1.014^{(0.12)}$ | $1.002^{(0.04)}$ | $1.027^{(0.16)}$ |
| RF | Score | $0.857^{(0.17)}$ | $0.870^{(0.17)}$ | $0.874^{(0.17)}$ | $1.249^{(0.43)}$ | $1.150^{(0.36)}$ | $1.110^{(0.31)}$ |
| | Rank | $0.855^{(0.18)}$ | $0.873^{(0.17)}$ | $0.874^{(0.17)}$ | $1.174^{(0.38)}$ | $1.069^{(0.25)}$ | $1.054^{(0.23)}$ |
| GB | Score | $0.852^{(0.18)}$ | $0.869^{(0.17)}$ | $0.872^{(0.17)}$ | $1.272^{(0.45)}$ | $1.089^{(0.28)}$ | $1.117^{(0.32)}$ |
| | Rank | $0.853^{(0.18)}$ | $0.871^{(0.17)}$ | $0.874^{(0.17)}$ | $1.240^{(0.43)}$ | $1.043^{(0.20)}$ | $1.034^{(0.18)}$ |
| LGBM | Score | $0.850^{(0.18)}$ | $0.870^{(0.17)}$ | $0.874^{(0.17)}$ | $1.416^{(0.49)}$ | $1.237^{(0.43)}$ | $1.131^{(0.34)}$ |
| | Rank | $0.855^{(0.18)}$ | $0.873^{(0.17)}$ | $0.875^{(0.17)}$ | $1.214^{(0.41)}$ | $1.081^{(0.27)}$ | $1.061^{(0.24)}$ |
| BO | Score | $0.854^{(0.18)}$ | $0.869^{(0.17)}$ | $0.873^{(0.17)}$ | $1.039^{(0.19)}$ | $1.006^{(0.08)}$ | $1.109^{(0.31)}$ |
| | Rank | $0.856^{(0.18)}$ | $0.869^{(0.17)}$ | $0.874^{(0.17)}$ | $1.016^{(0.13)}$ | $1.002^{(0.04)}$ | $1.026^{(0.16)}$ |

Table 5: Improvement Percentage: Rank-based over Score-based approaches

|  | Avg | LR | Lasso | Ridge | RF | GB | LGBM | BO |
|---|---|---|---|---|---|---|---|---|
| NDCG@1 | 0.008 | 0.008 | 0.269 | 0.008 | 0.016 | 0.025 | 0.059 | 0.007 |
| NDCG@5 | 0.002 | 0.002 | 0.227 | 0.002 | 0.013 | 0.014 | 0.038 | 0.002 |
| NDCG@10 | 0.003 | 0.003 | 0.192 | 0.003 | 0.017 | 0.011 | 0.030 | 0.002 |
| MRR@1 | 0.006 | 0.006 | 0.057 | 0.006 | 0.043 | -0.045 | 0.044 | 0.006 |
| MRR@5 | 0.003 | 0.003 | 0.160 | 0.003 | 0.044 | -0.012 | 0.076 | 0.003 |
| MRR@10 | 0.012 | 0.012 | 0.166 | 0.012 | 0.045 | -0.008 | 0.069 | 0.012 |
| SCORE@1 | 0.002 | 0.002 | 0.089 | 0.002 | -0.002 | 0.001 | 0.005 | 0.002 |
| SCORE@5 | 0.000 | 0.000 | 0.015 | 0.000 | 0.003 | 0.001 | 0.003 | 0.000 |
| SCORE@10 | 0.001 | 0.001 | 0.008 | 0.001 | 0.000 | 0.001 | 0.001 | 0.001 |
| RANK@1 | -0.027 | -0.027 | -0.638 | -0.027 | -0.074 | -0.032 | -0.202 | -0.023 |
| RANK@5 | -0.004 | -0.004 | -0.495 | -0.004 | -0.081 | -0.046 | -0.156 | -0.004 |
| RANK@10 | -0.082 | -0.082 | -0.421 | -0.082 | -0.056 | -0.083 | -0.070 | -0.083 |

Table 6: Results of the Wilcoxon signed rank test comparing the rankings induced by the ranked-based vs scored-based strategies. The statistically significant differences ($p\_value < 0.05$) in each group are highlighted in blue.

|  | Avg | LR | Lasso | Ridge | RF | GB | LGBM | BO |
|---|---|---|---|---|---|---|---|---|
| NDCG@1 | 0.000 | 0.000 | 0.000 | 0.000 | 0.000 | 0.000 | 0.000 | 0.000 |
| NDCG@5 | 0.000 | 0.000 | 0.000 | 0.000 | 0.000 | 0.000 | 0.000 | 0.000 |
| NDCG@10 | 0.000 | 0.000 | 0.000 | 0.000 | 0.000 | 0.000 | 0.000 | 0.000 |
| MRR@1 | 0.042 | 0.042 | 0.000 | 0.042 | 0.000 | 1.000 | 0.001 | 0.042 |
| MRR@5 | 0.083 | 0.114 | 0.000 | 0.083 | 0.000 | 0.945 | 0.000 | 0.083 |
| MRR@10 | 0.000 | 0.000 | 0.000 | 0.000 | 0.000 | 0.267 | 0.000 | 0.000 |
| SCORE@1 | 0.000 | 0.000 | 0.000 | 0.000 | 0.000 | 0.004 | 0.000 | 0.000 |
| SCORE@5 | 0.046 | 0.046 | 0.000 | 0.046 | 0.000 | 0.000 | 0.000 | 0.046 |
| SCORE@10 | 0.000 | 0.000 | 0.000 | 0.000 | 0.000 | 0.000 | 0.000 | 0.000 |
| RANK@1 | 0.000 | 0.000 | 0.000 | 0.000 | 0.000 | 0.071 | 0.000 | 0.001 |
| RANK@5 | 0.079 | 0.079 | 0.000 | 0.079 | 0.000 | 0.000 | 0.000 | 0.079 |
| RANK@10 | 0.000 | 0.000 | 0.000 | 0.000 | 0.000 | 0.000 | 0.000 | 0.000 |

## B.2 AMLB-CLASSIFICATION

Table 7: Comparison of approaches using ranking metrics NDCG and MRR on AMLB clasification tasks. The best results in each group are highlighted in bold.

|  | var | NDCG@1 | NDCG@10 | NDCG@100 | MRR@1 | MRR@10 | MRR@100 |
|---|---|---|---|---|---|---|---|
| Random |  | $0.679^{(0.23)}$ | $0.669^{(0.14)}$ | $0.686^{(0.12)}$ | $0.003^{(0.05)}$ | $0.009^{(0.06)}$ | $0.013^{(0.07)}$ |
| Avg | Score | $0.850^{(0.16)}$ | $0.859^{(0.13)}$ | $0.872^{(0.09)}$ | $0.001^{(0.04)}$ | $0.028^{(0.09)}$ | $0.035^{(0.09)}$ |
|  | Rank | $0.884^{(0.12)}$ | $0.890^{(0.10)}$ | $0.878^{(0.08)}$ | $0.023^{(0.15)}$ | $0.040^{(0.16)}$ | $0.049^{(0.16)}$ |
| LR | Score | $0.850^{(0.16)}$ | $0.860^{(0.13)}$ | $0.872^{(0.09)}$ | $0.001^{(0.04)}$ | $0.028^{(0.09)}$ | $0.035^{(0.09)}$ |
|  | Rank | $0.885^{(0.12)}$ | $0.890^{(0.10)}$ | $0.878^{(0.08)}$ | $0.023^{(0.15)}$ | $0.041^{(0.17)}$ | $0.049^{(0.16)}$ |
| Lasso | Score | $0.653^{(0.23)}$ | $0.662^{(0.14)}$ | $0.684^{(0.12)}$ | $0.003^{(0.05)}$ | $0.009^{(0.07)}$ | $0.013^{(0.07)}$ |
|  | Rank | $0.884^{(0.12)}$ | $0.890^{(0.10)}$ | $0.878^{(0.08)}$ | $0.023^{(0.15)}$ | $0.039^{(0.16)}$ | $0.048^{(0.16)}$ |
| Ridge | Score | $0.850^{(0.16)}$ | $0.860^{(0.13)}$ | $0.872^{(0.09)}$ | $0.001^{(0.04)}$ | $0.028^{(0.09)}$ | $0.035^{(0.09)}$ |
|  | Rank | $0.885^{(0.12)}$ | $0.890^{(0.10)}$ | $0.878^{(0.08)}$ | $0.023^{(0.15)}$ | $0.041^{(0.17)}$ | $0.049^{(0.16)}$ |
| RF | Score | $0.846^{(0.18)}$ | $0.845^{(0.14)}$ | $0.847^{(0.11)}$ | $0.044^{(0.20)}$ | $0.067^{(0.22)}$ | $0.075^{(0.22)}$ |
|  | Rank | $0.878^{(0.14)}$ | $0.876^{(0.11)}$ | $0.870^{(0.09)}$ | $0.030^{(0.17)}$ | $0.047^{(0.18)}$ | $0.056^{(0.18)}$ |
| GB | Score | $0.870^{(0.14)}$ | $0.872^{(0.11)}$ | $0.872^{(0.09)}$ | $0.023^{(0.15)}$ | $0.044^{(0.16)}$ | $0.051^{(0.16)}$ |
|  | Rank | $0.893^{(0.12)}$ | $0.894^{(0.09)}$ | $0.878^{(0.07)}$ | $0.023^{(0.15)}$ | $0.041^{(0.17)}$ | $0.050^{(0.17)}$ |
| LGBM | Score | $0.868^{(0.15)}$ | $0.870^{(0.11)}$ | $0.867^{(0.09)}$ | $0.015^{(0.12)}$ | $0.038^{(0.15)}$ | $0.045^{(0.15)}$ |
|  | Rank | $0.888^{(0.12)}$ | $0.885^{(0.09)}$ | $0.874^{(0.08)}$ | $0.021^{(0.14)}$ | $0.039^{(0.16)}$ | $0.047^{(0.16)}$ |
| BO | Score | $0.850^{(0.16)}$ | $0.860^{(0.13)}$ | $0.872^{(0.09)}$ | $0.001^{(0.04)}$ | $0.028^{(0.09)}$ | $0.035^{(0.09)}$ |
|  | Rank | $0.884^{(0.12)}$ | $0.890^{(0.10)}$ | $0.878^{(0.08)}$ | $0.023^{(0.15)}$ | $0.041^{(0.17)}$ | $0.049^{(0.16)}$ |
| MCTS | Score | $0.864^{(0.15)}$ | $0.855^{(0.13)}$ | $0.837^{(0.10)}$ | $0.030^{(0.17)}$ | $0.052^{(0.18)}$ | $0.061^{(0.18)}$ |
|  | Rank | $0.876^{(0.13)}$ | $0.852^{(0.10)}$ | $0.824^{(0.08)}$ | $0.051^{(0.22)}$ | $0.059^{(0.22)}$ | $0.068^{(0.22)}$ |

Table 8: Comparison of approaches using AutoML metrics average SCORE and average RANK on AMLB classification tasks. The best results in each group are highlighted in bold.

| | var | SCORE@1 | SCORE@10 | SCORE@100 | RANK@1 | RANK@10 | RANK@100 |
|---|---|---|---|---|---|---|---|
| Random | | $0.558^{(0.22)}$ | $0.723^{(0.18)}$ | $0.777^{(0.17)}$ | - | - | - |
| Avg | Score | $0.675^{(0.20)}$ | $0.723^{(0.19)}$ | $0.767^{(0.17)}$ | $1.500^{(0.50)}$ | $1.238^{(0.43)}$ | $1.172^{(0.38)}$ |
| | Rank | $0.701^{(0.18)}$ | $0.720^{(0.18)}$ | $0.772^{(0.17)}$ | $1.366^{(0.48)}$ | $1.399^{(0.49)}$ | $1.172^{(0.38)}$ |
| LR | Score | $0.675^{(0.20)}$ | $0.723^{(0.19)}$ | $0.767^{(0.17)}$ | $1.504^{(0.50)}$ | $1.238^{(0.43)}$ | $1.170^{(0.38)}$ |
| | Rank | $0.702^{(0.18)}$ | $0.720^{(0.18)}$ | $0.772^{(0.17)}$ | $1.361^{(0.48)}$ | $1.392^{(0.49)}$ | $1.168^{(0.37)}$ |
| Lasso | Score | $0.545^{(0.22)}$ | $0.724^{(0.18)}$ | $0.776^{(0.17)}$ | $1.813^{(0.39)}$ | $1.469^{(0.50)}$ | $1.510^{(0.50)}$ |
| | Rank | $0.701^{(0.18)}$ | $0.720^{(0.18)}$ | $0.772^{(0.17)}$ | $1.156^{(0.36)}$ | $1.473^{(0.50)}$ | $1.331^{(0.47)}$ |
| Ridge | Score | $0.675^{(0.20)}$ | $0.723^{(0.19)}$ | $0.767^{(0.17)}$ | $1.504^{(0.50)}$ | $1.238^{(0.43)}$ | $1.170^{(0.38)}$ |
| | Rank | $0.702^{(0.18)}$ | $0.720^{(0.18)}$ | $0.772^{(0.17)}$ | $1.361^{(0.48)}$ | $1.392^{(0.49)}$ | $1.168^{(0.37)}$ |
| RF | Score | $0.674^{(0.19)}$ | $0.739^{(0.17)}$ | $0.779^{(0.17)}$ | $1.485^{(0.50)}$ | $1.361^{(0.48)}$ | $1.306^{(0.46)}$ |
| | Rank | $0.688^{(0.19)}$ | $0.737^{(0.18)}$ | $0.781^{(0.17)}$ | $1.427^{(0.49)}$ | $1.437^{(0.50)}$ | $1.272^{(0.44)}$ |
| GB | Score | $0.684^{(0.20)}$ | $0.734^{(0.18)}$ | $0.768^{(0.17)}$ | $1.459^{(0.50)}$ | $1.282^{(0.45)}$ | $1.193^{(0.39)}$ |
| | Rank | $0.702^{(0.18)}$ | $0.731^{(0.18)}$ | $0.772^{(0.17)}$ | $1.375^{(0.48)}$ | $1.377^{(0.48)}$ | $1.199^{(0.40)}$ |
| LGBM | Score | $0.688^{(0.19)}$ | $0.735^{(0.18)}$ | $0.770^{(0.17)}$ | $1.423^{(0.49)}$ | $1.327^{(0.47)}$ | $1.225^{(0.42)}$ |
| | Rank | $0.697^{(0.18)}$ | $0.734^{(0.18)}$ | $0.773^{(0.17)}$ | $1.420^{(0.49)}$ | $1.377^{(0.48)}$ | $1.208^{(0.41)}$ |
| BO | Score | $0.675^{(0.20)}$ | $0.723^{(0.19)}$ | $0.767^{(0.17)}$ | $1.503^{(0.50)}$ | $1.238^{(0.43)}$ | $1.170^{(0.38)}$ |
| | Rank | $0.701^{(0.18)}$ | $0.720^{(0.18)}$ | $0.772^{(0.17)}$ | $1.358^{(0.48)}$ | $1.392^{(0.49)}$ | $1.168^{(0.37)}$ |
| MCTS | Score | $0.686^{(0.19)}$ | $0.725^{(0.18)}$ | $0.774^{(0.17)}$ | $1.306^{(0.46)}$ | $1.454^{(0.50)}$ | $1.286^{(0.45)}$ |
| | Rank | $0.698^{(0.18)}$ | $0.733^{(0.18)}$ | $0.773^{(0.17)}$ | $1.325^{(0.47)}$ | $1.368^{(0.48)}$ | $1.341^{(0.47)}$ |

Table 9: Comparison of approaches using average TTB on AMLB classification tasks. The best results in each group are highlighted in bold.

| | var | TTB@1 | TTB@10 | TTB@100 |
|---|---|---|---|---|
| Random | | $5041.482^{(3583.41)}$ | $4908.858^{(3639.28)}$ | $4081.935^{(3736.80)}$ |
| Avg | Score | $5043.765^{(3580.72)}$ | $4448.601^{(3677.92)}$ | $3033.816^{(3525.23)}$ |
| | Rank | $4874.958^{(3624.69)}$ | $4362.415^{(3660.09)}$ | $2926.108^{(3514.38)}$ |
| LR | Score | $5043.765^{(3580.72)}$ | $4437.394^{(3679.17)}$ | $3033.332^{(3524.90)}$ |
| | Rank | $4874.908^{(3624.76)}$ | $4352.566^{(3662.46)}$ | $2924.392^{(3515.49)}$ |
| Lasso | Score | $5032.579^{(3583.86)}$ | $4876.417^{(3636.97)}$ | $4203.129^{(3747.74)}$ |
| | Rank | $4874.958^{(3624.69)}$ | $4362.433^{(3660.07)}$ | $2925.991^{(3514.42)}$ |
| Ridge | Score | $5043.765^{(3580.72)}$ | $4437.394^{(3679.17)}$ | $3033.332^{(3524.90)}$ |
| | Rank | $4874.908^{(3624.76)}$ | $4352.566^{(3662.46)}$ | $2924.392^{(3515.49)}$ |
| RF | Score | $4787.428^{(3668.17)}$ | $4249.099^{(3724.16)}$ | $2849.469^{(3502.55)}$ |
| | Rank | $4934.319^{(3615.37)}$ | $4578.992^{(3718.60)}$ | $2939.404^{(3455.19)}$ |
| GB | Score | $4927.551^{(3616.19)}$ | $4347.207^{(3679.70)}$ | $3061.660^{(3518.02)}$ |
| | Rank | $4873.310^{(3623.17)}$ | $4367.823^{(3672.12)}$ | $2988.906^{(3558.88)}$ |
| LGBM | Score | $4947.902^{(3602.84)}$ | $4509.409^{(3674.22)}$ | $3138.629^{(3504.85)}$ |
| | Rank | $4878.782^{(3613.86)}$ | $4369.698^{(3668.37)}$ | $3075.300^{(3562.53)}$ |
| BO | Score | $5043.765^{(3580.72)}$ | $4437.394^{(3679.17)}$ | $3033.415^{(3524.88)}$ |
| | Rank | $4874.908^{(3624.76)}$ | $4352.566^{(3662.46)}$ | $2924.351^{(3515.50)}$ |
| MCTS | Score | $4812.553^{(3628.41)}$ | $4218.461^{(3735.58)}$ | $2965.364^{(3726.11)}$ |
| | Rank | $4631.098^{(3651.16)}$ | $4330.997^{(3729.38)}$ | $3165.093^{(3672.53)}$ |

Table 10: Improvement Percentage: Rank-based over Score-based approaches

|  | Avg | LR | Lasso | Ridge | RF | GB | LGBM | BO | MCTS |
|---|---|---|---|---|---|---|---|---|---|
| NDCG@1 | 0.040 | 0.041 | 0.353 | 0.041 | 0.038 | 0.027 | 0.023 | 0.041 | 0.014 |
| NDCG@10 | 0.036 | 0.036 | 0.344 | 0.036 | 0.037 | 0.025 | 0.017 | 0.036 | -0.003 |
| NDCG@100 | 0.007 | 0.007 | 0.284 | 0.007 | 0.027 | 0.007 | 0.008 | 0.007 | -0.016 |
| MRR@1 | 15.000 | 15.000 | 7.000 | 15.000 | -0.323 | 0.000 | 0.364 | 15.000 | 0.714 |
| MRR@10 | 0.430 | 0.457 | 3.446 | 0.457 | -0.297 | -0.061 | 0.032 | 0.457 | 0.145 |
| MRR@100 | 0.374 | 0.398 | 2.641 | 0.398 | -0.260 | -0.027 | 0.036 | 0.398 | 0.110 |
| SCORE@1 | 0.039 | 0.040 | 0.287 | 0.040 | 0.021 | 0.026 | 0.013 | 0.040 | 0.017 |
| SCORE@10 | -0.005 | -0.004 | -0.007 | -0.004 | -0.003 | -0.004 | -0.002 | -0.005 | 0.011 |
| SCORE@100 | 0.007 | 0.007 | -0.005 | 0.007 | 0.003 | 0.006 | 0.004 | 0.007 | -0.001 |
| TTB@1 | -0.033 | -0.033 | -0.031 | -0.033 | 0.031 | -0.011 | -0.014 | -0.033 | -0.038 |
| TTB@10 | -0.019 | -0.019 | -0.105 | -0.019 | 0.078 | 0.005 | -0.031 | -0.019 | 0.027 |
| TTB@100 | -0.036 | -0.036 | -0.304 | -0.036 | 0.032 | -0.024 | -0.020 | -0.036 | 0.067 |
| RANK@1 | -0.089 | -0.096 | -0.362 | -0.096 | -0.039 | -0.058 | -0.002 | -0.097 | 0.015 |
| RANK@10 | 0.130 | 0.124 | 0.003 | 0.124 | 0.056 | 0.075 | 0.038 | 0.124 | -0.059 |
| RANK@100 | 0.000 | -0.002 | -0.118 | -0.002 | -0.026 | 0.005 | -0.014 | -0.002 | 0.043 |

Table 11: Results of the Wilcoxon signed rank test comparing the rankings induced by the ranked-based vs scored-based strategies. The statistically significant differences (*p_value* $< 0.05$) in each group are highlighted in blue.

|  | Avg | LR | Lasso | Ridge | RF | GB | LGBM | BO | MCTS |
|---|---|---|---|---|---|---|---|---|---|
| NDCG@1 | 0.000 | 0.000 | 0.000 | 0.000 | 0.001 | 0.001 | 0.008 | 0.000 | 0.137 |
| NDCG@10 | 0.000 | 0.000 | 0.000 | 0.000 | 0.000 | 0.000 | 0.000 | 0.000 | 0.992 |
| NDCG@100 | 0.004 | 0.003 | 0.000 | 0.003 | 0.000 | 0.002 | 0.000 | 0.003 | 1.000 |
| MRR@1 | 0.000 | 0.000 | 0.000 | 0.000 | 0.926 | 0.500 | 0.197 | 0.000 | 0.004 |
| MRR@10 | 0.108 | 0.081 | 0.000 | 0.081 | 0.981 | 0.618 | 0.411 | 0.081 | 0.356 |
| MRR@100 | 0.006 | 0.005 | 0.000 | 0.005 | 0.943 | 0.462 | 0.252 | 0.005 | 0.995 |
| SCORE@1 | 0.000 | 0.000 | 0.000 | 0.000 | 0.007 | 0.002 | 0.107 | 0.000 | 0.359 |
| SCORE@10 | 1.000 | 1.000 | 0.843 | 1.000 | 0.999 | 1.000 | 0.930 | 1.000 | 0.000 |
| SCORE@100 | 0.001 | 0.001 | 0.144 | 0.001 | 0.022 | 0.012 | 0.018 | 0.001 | 0.578 |
| TTB@1 | 0.000 | 0.000 | 0.000 | 0.000 | 0.990 | 0.258 | 0.081 | 0.000 | 0.000 |
| TTB@10 | 0.945 | 0.933 | 0.000 | 0.933 | 1.000 | 0.974 | 0.338 | 0.933 | 0.978 |
| TTB@100 | 0.000 | 0.000 | 0.000 | 0.000 | 0.971 | 0.001 | 0.006 | 0.000 | 0.863 |
| AVG_RANK@1 | 0.000 | 0.000 | 0.000 | 0.000 | 0.053 | 0.007 | 0.467 | 0.000 | 0.746 |
| AVG_RANK@10 | 1.000 | 1.000 | 0.546 | 1.000 | 0.988 | 0.999 | 0.946 | 1.000 | 0.006 |
| AVG_RANK@100 | 0.500 | 0.449 | 0.000 | 0.449 | 0.118 | 0.595 | 0.247 | 0.449 | 0.968 |

## B.3 AMLB-REGRESSION

Table 12: Comparison of approaches using ranking metrics NDCG and MRR on AMLB regression tasks. The best results in each group are highlighted in bold.

| | var | NDCG@1 | NDCG@10 | NDCG@100 | MRR@1 | MRR@10 | MRR@100 |
|---|---|---|---|---|---|---|---|
| Random | | $0.711^{(0.20)}$ | $0.715^{(0.12)}$ | $0.736^{(0.11)}$ | $0.000^{(0.00)}$ | $0.010^{(0.06)}$ | $0.016^{(0.06)}$ |
| Avg | Score | $0.778^{(0.23)}$ | $0.789^{(0.16)}$ | $0.817^{(0.10)}$ | $0.039^{(0.19)}$ | $0.062^{(0.20)}$ | $0.068^{(0.20)}$ |
| | Rank | $0.846^{(0.19)}$ | $0.855^{(0.15)}$ | $0.868^{(0.11)}$ | $0.073^{(0.26)}$ | $0.113^{(0.28)}$ | $0.123^{(0.28)}$ |
| LR | Score | $0.723^{(0.21)}$ | $0.715^{(0.13)}$ | $0.736^{(0.11)}$ | $0.012^{(0.11)}$ | $0.016^{(0.11)}$ | $0.022^{(0.11)}$ |
| | Rank | $0.845^{(0.20)}$ | $0.855^{(0.15)}$ | $0.869^{(0.11)}$ | $0.079^{(0.27)}$ | $0.116^{(0.29)}$ | $0.125^{(0.28)}$ |
| Lasso | Score | $0.721^{(0.22)}$ | $0.714^{(0.13)}$ | $0.736^{(0.11)}$ | $0.012^{(0.11)}$ | $0.016^{(0.11)}$ | $0.022^{(0.11)}$ |
| | Rank | $0.846^{(0.19)}$ | $0.855^{(0.15)}$ | $0.869^{(0.11)}$ | $0.073^{(0.26)}$ | $0.113^{(0.28)}$ | $0.123^{(0.28)}$ |
| Ridge | Score | $0.721^{(0.22)}$ | $0.714^{(0.13)}$ | $0.736^{(0.11)}$ | $0.012^{(0.11)}$ | $0.016^{(0.11)}$ | $0.022^{(0.11)}$ |
| | Rank | $0.845^{(0.20)}$ | $0.855^{(0.15)}$ | $0.869^{(0.11)}$ | $0.079^{(0.27)}$ | $0.116^{(0.29)}$ | $0.125^{(0.28)}$ |
| RF | Score | $0.702^{(0.22)}$ | $0.711^{(0.13)}$ | $0.737^{(0.11)}$ | $0.000^{(0.00)}$ | $0.009^{(0.05)}$ | $0.014^{(0.05)}$ |
| | Rank | $0.858^{(0.17)}$ | $0.858^{(0.15)}$ | $0.863^{(0.13)}$ | $0.030^{(0.17)}$ | $0.062^{(0.19)}$ | $0.075^{(0.19)}$ |
| GB | Score | $0.693^{(0.23)}$ | $0.708^{(0.13)}$ | $0.734^{(0.11)}$ | $0.003^{(0.05)}$ | $0.008^{(0.06)}$ | $0.014^{(0.06)}$ |
| | Rank | $0.835^{(0.20)}$ | $0.847^{(0.16)}$ | $0.868^{(0.11)}$ | $0.067^{(0.25)}$ | $0.093^{(0.26)}$ | $0.105^{(0.26)}$ |
| LGBM | Score | $0.613^{(0.26)}$ | $0.663^{(0.18)}$ | $0.724^{(0.12)}$ | $0.000^{(0.00)}$ | $0.006^{(0.03)}$ | $0.012^{(0.03)}$ |
| | Rank | $0.828^{(0.20)}$ | $0.839^{(0.16)}$ | $0.859^{(0.12)}$ | $0.055^{(0.23)}$ | $0.076^{(0.24)}$ | $0.089^{(0.23)}$ |
| BO | Score | $0.796^{(0.21)}$ | $0.804^{(0.14)}$ | $0.816^{(0.10)}$ | $0.027^{(0.16)}$ | $0.062^{(0.19)}$ | $0.070^{(0.19)}$ |
| | Rank | $0.846^{(0.19)}$ | $0.855^{(0.15)}$ | $0.869^{(0.11)}$ | $0.079^{(0.27)}$ | $0.116^{(0.29)}$ | $0.125^{(0.28)}$ |
| MCTS | Score | $0.733^{(0.23)}$ | $0.732^{(0.19)}$ | $0.754^{(0.14)}$ | $0.052^{(0.22)}$ | $0.066^{(0.22)}$ | $0.072^{(0.22)}$ |
| | Rank | $0.878^{(0.17)}$ | $0.845^{(0.11)}$ | $0.851^{(0.09)}$ | $0.085^{(0.28)}$ | $0.085^{(0.28)}$ | $0.091^{(0.28)}$ |

Table 13: Comparison of approaches using AutoML metrics average SCORE and average RANK on AMLB regression tasks. The best results in each group are highlighted in bold.

| | var | SCORE@1 | SCORE@10 | SCORE@100 | RANK@1 | RANK@10 | RANK@100 |
|---|---|---|---|---|---|---|---|
| Random | | $-1.90e+22^{(3.02e+23)}$ | $-2.13e+06^{(1.06e+07)}$ | $-1.94e+06^{(9.89e+06)}$ | - | - | - |
| Avg | Score | $-2.28e+06^{(1.07e+07)}$ | $-2.01e+06^{(1.01e+07)}$ | $-1.90e+06^{(9.70e+06)}$ | $1.418^{(0.49)}$ | $1.403^{(0.49)}$ | $1.358^{(0.48)}$ |
| | Rank | $-2.10e+06^{(1.07e+07)}$ | $-1.91e+06^{(9.70e+06)}$ | $-1.91e+06^{(9.70e+06)}$ | $1.382^{(0.49)}$ | $1.352^{(0.48)}$ | $1.264^{(0.44)}$ |
| LR | Score | $-1.26e+14^{(2.27e+15)}$ | $-2.19e+06^{(1.10e+07)}$ | $-1.95e+06^{(9.95e+06)}$ | $1.721^{(0.45)}$ | $1.630^{(0.48)}$ | $1.464^{(0.50)}$ |
| | Rank | $-2.08e+06^{(1.06e+07)}$ | $-1.91e+06^{(9.70e+06)}$ | $-1.91e+06^{(9.70e+06)}$ | $1.267^{(0.44)}$ | $1.321^{(0.47)}$ | $1.339^{(0.47)}$ |
| Lasso | Score | $-1.26e+14^{(2.27e+15)}$ | $-2.19e+06^{(1.10e+07)}$ | $-1.95e+06^{(9.94e+06)}$ | $1.721^{(0.45)}$ | $1.627^{(0.48)}$ | $1.464^{(0.50)}$ |
| | Rank | $-2.10e+06^{(1.07e+07)}$ | $-1.91e+06^{(9.70e+06)}$ | $-1.91e+06^{(9.70e+06)}$ | $1.267^{(0.44)}$ | $1.327^{(0.47)}$ | $1.339^{(0.47)}$ |
| Ridge | Score | $-1.26e+14^{(2.27e+15)}$ | $-2.19e+06^{(1.10e+07)}$ | $-1.95e+06^{(9.94e+06)}$ | $1.721^{(0.45)}$ | $1.633^{(0.48)}$ | $1.467^{(0.50)}$ |
| | Rank | $-2.08e+06^{(1.06e+07)}$ | $-1.91e+06^{(9.70e+06)}$ | $-1.91e+06^{(9.70e+06)}$ | $1.267^{(0.44)}$ | $1.324^{(0.47)}$ | $1.339^{(0.47)}$ |
| RF | Score | $-2.19e+37^{(3.97e+38)}$ | $-2.11e+06^{(1.06e+07)}$ | $-1.94e+06^{(9.92e+06)}$ | $1.782^{(0.41)}$ | $1.506^{(0.50)}$ | $1.424^{(0.49)}$ |
| | Rank | $-2.34e+06^{(1.17e+07)}$ | $-2.23e+06^{(1.14e+07)}$ | $-1.98e+06^{(1.01e+07)}$ | $1.215^{(0.41)}$ | $1.467^{(0.50)}$ | $1.379^{(0.49)}$ |
| GB | Score | $-3.90e+58^{(2.21e+59)}$ | $-2.13e+06^{(1.07e+07)}$ | $-1.95e+06^{(9.91e+06)}$ | $1.742^{(0.44)}$ | $1.633^{(0.48)}$ | $1.521^{(0.50)}$ |
| | Rank | $-2.09e+06^{(1.06e+07)}$ | $-1.97e+06^{(1.00e+07)}$ | $-1.91e+06^{(9.72e+06)}$ | $1.242^{(0.43)}$ | $1.309^{(0.46)}$ | $1.279^{(0.45)}$ |
| LGBM | Score | $-6.71e+57^{(4.93e+58)}$ | $-1.40e+34^{(1.33e+35)}$ | $-1.93e+06^{(9.82e+06)}$ | $1.833^{(0.37)}$ | $1.655^{(0.48)}$ | $1.439^{(0.50)}$ |
| | Rank | $-5.53e+06^{(6.37e+07)}$ | $-1.98e+06^{(1.01e+07)}$ | $-1.92e+06^{(9.78e+06)}$ | $1.164^{(0.37)}$ | $1.309^{(0.46)}$ | $1.306^{(0.46)}$ |
| BO | Score | $-2.39e+06^{(1.17e+07)}$ | $-2.01e+06^{(1.01e+07)}$ | $-1.91e+06^{(9.74e+06)}$ | $1.542^{(0.50)}$ | $1.436^{(0.50)}$ | $1.367^{(0.48)}$ |
| | Rank | $-2.08e+06^{(1.06e+07)}$ | $-1.91e+06^{(9.70e+06)}$ | $-1.91e+06^{(9.70e+06)}$ | $1.370^{(0.48)}$ | $1.367^{(0.48)}$ | $1.273^{(0.45)}$ |
| MCTS | Score | $-2.31e+06^{(1.08e+07)}$ | $-2.13e+06^{(1.01e+07)}$ | $-1.92e+06^{(9.70e+06)}$ | $1.639^{(0.48)}$ | $1.597^{(0.49)}$ | $1.379^{(0.49)}$ |
| | Rank | $-1.92e+06^{(9.70e+06)}$ | $-1.91e+06^{(9.70e+06)}$ | $-1.91e+06^{(9.70e+06)}$ | $1.330^{(0.47)}$ | $1.348^{(0.48)}$ | $1.309^{(0.46)}$ |

Table 14: Comparison of approaches using average TTB on AMLB regression tasks. The best results in each group are highlighted in bold.

| | var | TTB@1 | TTB@10 | TTB@100 |
|---|---|---|---|---|
| Random | | $3586.084^{(1832.81)}$ | $3453.575^{(1911.90)}$ | $2761.787^{(2145.84)}$ |
| Avg | Score | $3424.625^{(1951.65)}$ | $2903.705^{(2142.14)}$ | $2111.220^{(1812.69)}$ |
| | Rank | $3290.290^{(2042.10)}$ | $2708.923^{(2060.88)}$ | $1859.865^{(1852.73)}$ |
| LR | Score | $3545.273^{(1864.66)}$ | $3455.948^{(1928.41)}$ | $2747.323^{(2190.03)}$ |
| | Rank | $3255.857^{(2045.68)}$ | $2690.243^{(2056.09)}$ | $1859.944^{(1852.76)}$ |
| Lasso | Score | $3545.273^{(1864.66)}$ | $3455.916^{(1928.47)}$ | $2729.861^{(2181.15)}$ |
| | Rank | $3290.291^{(2042.10)}$ | $2708.788^{(2061.04)}$ | $1859.427^{(1852.95)}$ |
| Ridge | Score | $3545.273^{(1864.66)}$ | $3455.916^{(1928.47)}$ | $2737.503^{(2177.95)}$ |
| | Rank | $3255.857^{(2045.68)}$ | $2690.243^{(2056.09)}$ | $1859.944^{(1852.76)}$ |
| RF | Score | $3586.084^{(1832.81)}$ | $3458.212^{(1897.20)}$ | $2843.086^{(2124.43)}$ |
| | Rank | $3477.630^{(1924.12)}$ | $3024.837^{(2140.76)}$ | $1941.372^{(2036.63)}$ |
| GB | Score | $3572.849^{(1842.57)}$ | $3485.569^{(1903.98)}$ | $2798.697^{(2151.61)}$ |
| | Rank | $3317.121^{(2027.53)}$ | $2849.755^{(2131.73)}$ | $1731.679^{(1866.51)}$ |
| LGBM | Score | $3586.084^{(1832.81)}$ | $3416.294^{(1896.80)}$ | $2541.332^{(2095.30)}$ |
| | Rank | $3371.637^{(1979.92)}$ | $2941.128^{(2091.28)}$ | $1757.136^{(1800.78)}$ |
| BO | Score | $3477.129^{(1919.76)}$ | $2986.739^{(2124.44)}$ | $2175.204^{(1870.60)}$ |
| | Rank | $3255.857^{(2045.68)}$ | $2690.243^{(2056.08)}$ | $1859.945^{(1852.76)}$ |
| MCTS | Score | $3378.009^{(1983.92)}$ | $3145.425^{(2127.41)}$ | $2319.453^{(2258.28)}$ |
| | Rank | $3148.875^{(1956.02)}$ | $3137.958^{(1963.27)}$ | $2549.071^{(2148.62)}$ |

Table 15: Improvement Percentage: Rank-based over Score-based approaches

| | Avg | LR | Lasso | Ridge | RF | GB | LGBM | BO | MCTS |
|---|---|---|---|---|---|---|---|---|---|
| NDCG@1 | 0.088 | 0.169 | 0.173 | 0.173 | 0.221 | 0.204 | 0.351 | 0.063 | 0.198 |
| NDCG@10 | 0.083 | 0.196 | 0.198 | 0.198 | 0.207 | 0.195 | 0.266 | 0.064 | 0.154 |
| NDCG@100 | 0.063 | 0.180 | 0.180 | 0.180 | 0.171 | 0.183 | 0.186 | 0.065 | 0.129 |
| MRR@1 | 0.846 | 5.500 | 5.000 | 5.500 | | 21.000 | | 1.889 | 0.647 |
| MRR@10 | 0.826 | 6.433 | 6.107 | 6.282 | 6.233 | 10.791 | 12.154 | 0.862 | 0.292 |
| MRR@100 | 0.791 | 4.689 | 4.466 | 4.592 | 4.310 | 6.432 | 6.370 | 0.785 | 0.258 |
| SCORE@1 | -8.08e-02 | -1.00e+00 | -1.00e+00 | -1.00e+00 | -1.00e+00 | -1.00e+00 | -1.00e+00 | -1.28e-01 | -1.70e-01 |
| SCORE@10 | -4.60e-02 | -1.27e-01 | -1.27e-01 | -1.27e-01 | 5.83e-02 | -7.56e-02 | -1.00e+00 | -4.69e-02 | -1.01e-01 |
| SCORE@100 | 2.48e-03 | -2.13e-02 | -1.98e-02 | -1.98e-02 | 2.00e-02 | -1.81e-02 | -4.64e-03 | -6.35e-04 | -3.94e-03 |
| TTB@1 | -0.039 | -0.082 | -0.072 | -0.082 | -0.030 | -0.072 | -0.060 | -0.064 | -0.068 |
| TTB@10 | -0.067 | -0.222 | -0.216 | -0.222 | -0.125 | -0.182 | -0.139 | -0.099 | -0.002 |
| TTB@100 | -0.119 | -0.323 | -0.319 | -0.321 | -0.317 | -0.381 | -0.309 | -0.145 | 0.099 |
| RANK@1 | -0.026 | -0.264 | -0.264 | -0.264 | -0.318 | -0.287 | -0.365 | -0.112 | -0.189 |
| RANK@10 | -0.037 | -0.190 | -0.184 | -0.189 | -0.026 | -0.199 | -0.209 | -0.049 | -0.156 |
| RANK@100 | -0.069 | -0.085 | -0.085 | -0.087 | -0.032 | -0.159 | -0.093 | -0.069 | -0.051 |

Table 16: Results of the Wilcoxon signed rank test comparing the rankings induced by the ranked-based vs scored-based strategies. The statistically significant differences ($p\_value < 0.05$) in each group are highlighted in blue.

| | Avg | LR | Lasso | Ridge | RF | GB | LGBM | BO | MCTS |
|---|---|---|---|---|---|---|---|---|---|
| NDCG@1 | 0.001 | 0.000 | 0.000 | 0.000 | 0.000 | 0.000 | 0.000 | 0.000 | 0.000 |
| NDCG@10 | 0.000 | 0.000 | 0.000 | 0.000 | 0.000 | 0.000 | 0.000 | 0.000 | 0.000 |
| NDCG@100 | 0.000 | 0.000 | 0.000 | 0.000 | 0.000 | 0.000 | 0.000 | 0.000 | 0.000 |
| MRR@1 | 0.004 | 0.000 | 0.000 | 0.000 | 0.001 | 0.000 | 0.000 | 0.000 | 0.039 |
| MRR@10 | 0.000 | 0.000 | 0.000 | 0.000 | 0.000 | 0.000 | 0.000 | 0.000 | 0.304 |
| MRR@100 | 0.000 | 0.000 | 0.000 | 0.000 | 0.000 | 0.000 | 0.000 | 0.002 | 0.517 |
| SCORE@1 | 0.000 | 0.000 | 0.000 | 0.000 | 0.000 | 0.000 | 0.000 | 0.000 | 0.000 |
| SCORE@10 | 0.000 | 0.000 | 0.000 | 0.000 | 0.899 | 0.000 | 0.000 | 0.012 | 0.000 |
| SCORE@100 | 0.813 | 0.004 | 0.005 | 0.004 | 0.686 | 0.000 | 0.003 | 0.697 | 0.001 |
| TTB@1 | 0.250 | 0.000 | 0.000 | 0.000 | 0.006 | 0.000 | 0.000 | 0.011 | 0.000 |
| TTB@10 | 0.000 | 0.000 | 0.000 | 0.000 | 0.000 | 0.000 | 0.000 | 0.000 | 0.000 |
| TTB@100 | 0.000 | 0.000 | 0.000 | 0.000 | 0.000 | 0.000 | 0.000 | 0.000 | 0.072 |
| AVG_RANK@1 | 0.230 | 0.000 | 0.000 | 0.000 | 0.000 | 0.000 | 0.000 | 0.001 | 0.000 |
| AVG_RANK@10 | 0.141 | 0.000 | 0.000 | 0.000 | 0.234 | 0.000 | 0.000 | 0.079 | 0.000 |
| AVG_RANK@100 | 0.015 | 0.006 | 0.006 | 0.005 | 0.178 | 0.000 | 0.003 | 0.016 | 0.063 |

## B.4 CORRELATION BETWEEN METRICS

Tables 17, 18 and 19 show all Spearman correlations between all the metrics calculated for each dataset, respectively.

Table 17: Correlation between metrics in OpenML-Weka dataset.

| | N@1 | N@5 | N@10 | M@1 | M@5 | M@10 | S@1 | S@5 | S@10 | R@1 | R@5 | R@10 |
|---|---|---|---|---|---|---|---|---|---|---|---|---|
| N@1 | 1.00 | 0.87 | 0.72 | 0.57 | 0.57 | 0.55 | 0.88 | 0.55 | 0.86 | -0.41 | -0.27 | -0.56 |
| N@5 | 0.87 | 1.00 | 0.85 | 0.68 | 0.71 | 0.69 | 0.73 | 0.73 | 0.84 | -0.27 | -0.25 | -0.65 |
| N@10 | 0.72 | 0.85 | 1.00 | 0.81 | 0.81 | 0.78 | 0.55 | 0.86 | 0.80 | 0.01 | 0.05 | -0.39 |
| M@1 | 0.57 | 0.68 | 0.81 | 1.00 | 0.99 | 0.98 | 0.40 | 0.97 | 0.63 | 0.32 | 0.29 | -0.13 |
| M@5 | 0.57 | 0.71 | 0.81 | 0.99 | 1.00 | 0.99 | 0.37 | 0.97 | 0.66 | 0.28 | 0.21 | -0.21 |
| M@10 | 0.55 | 0.69 | 0.78 | 0.98 | 0.99 | 1.00 | 0.38 | 0.96 | 0.64 | 0.30 | 0.23 | -0.20 |
| S@1 | 0.88 | 0.73 | 0.55 | 0.40 | 0.37 | 0.38 | 1.00 | 0.35 | 0.74 | -0.60 | -0.43 | -0.61 |
| S@5 | 0.55 | 0.73 | 0.86 | 0.97 | 0.97 | 0.96 | 0.35 | 1.00 | 0.68 | 0.30 | 0.25 | -0.22 |
| S@10 | 0.86 | 0.84 | 0.80 | 0.63 | 0.66 | 0.64 | 0.74 | 0.68 | 1.00 | -0.36 | -0.28 | -0.74 |
| R@1 | -0.41 | -0.27 | 0.01 | 0.32 | 0.28 | 0.30 | -0.60 | 0.30 | -0.36 | 1.00 | 0.95 | 0.76 |
| R@5 | -0.27 | -0.25 | 0.05 | 0.29 | 0.21 | 0.23 | -0.43 | 0.25 | -0.28 | 0.95 | 1.00 | 0.78 |
| R@10 | -0.56 | -0.65 | -0.39 | -0.13 | -0.21 | -0.20 | -0.61 | -0.22 | -0.74 | 0.76 | 0.78 | 1.00 |

Table 18: Correlation between metrics in AMLB Classification dataset.

| | N@1 | N@10 | N@100 | M@1 | M@10 | M@100 | S@1 | S@10 | S@100 | T@1 | T@10 | T@100 | R@1 | R@10 | R@100 |
|---|---|---|---|---|---|---|---|---|---|---|---|---|---|---|---|
| N@1 | 1.00 | 0.90 | 0.76 | 0.40 | 0.42 | 0.42 | 0.95 | -0.20 | -0.05 | -0.51 | -0.36 | -0.41 | -0.67 | 0.36 | -0.28 |
| N@10 | 0.90 | 1.00 | 0.92 | 0.17 | 0.25 | 0.25 | 0.88 | -0.37 | -0.26 | -0.27 | -0.22 | -0.55 | -0.46 | 0.25 | -0.55 |
| N@100 | 0.76 | 0.92 | 1.00 | 0.03 | 0.13 | 0.13 | 0.75 | -0.59 | -0.39 | -0.17 | -0.31 | -0.63 | -0.36 | 0.12 | -0.72 |
| M@1 | 0.40 | 0.17 | 0.03 | 1.00 | 0.93 | 0.93 | 0.46 | 0.28 | 0.63 | -0.90 | -0.70 | -0.44 | -0.66 | 0.52 | 0.39 |
| M@10 | 0.42 | 0.25 | 0.13 | 0.93 | 1.00 | 1.00 | 0.46 | 0.36 | 0.37 | -0.84 | -0.79 | -0.50 | -0.57 | 0.28 | 0.24 |
| M@100 | 0.42 | 0.25 | 0.13 | 0.93 | 1.00 | 1.00 | 0.46 | 0.36 | 0.37 | -0.84 | -0.79 | -0.50 | -0.57 | 0.28 | 0.23 |
| S@1 | 0.95 | 0.88 | 0.75 | 0.46 | 0.46 | 0.46 | 1.00 | -0.32 | -0.06 | -0.58 | -0.42 | -0.49 | -0.78 | 0.42 | -0.28 |
| S@10 | -0.20 | -0.37 | -0.59 | 0.28 | 0.36 | 0.36 | -0.32 | 1.00 | 0.36 | -0.13 | 0.04 | 0.36 | 0.29 | -0.22 | 0.51 |
| S@100 | -0.05 | -0.26 | -0.39 | 0.63 | 0.37 | 0.37 | -0.06 | 0.36 | 1.00 | -0.49 | -0.10 | -0.07 | -0.35 | 0.75 | 0.64 |
| T@1 | -0.51 | -0.27 | -0.17 | -0.90 | -0.84 | -0.84 | -0.58 | -0.13 | -0.49 | 1.00 | 0.80 | 0.46 | 0.74 | -0.45 | -0.23 |
| T@10 | -0.36 | -0.22 | -0.31 | -0.70 | -0.79 | -0.79 | -0.42 | 0.04 | -0.10 | 0.80 | 1.00 | 0.53 | 0.62 | -0.14 | 0.02 |
| T@100 | -0.41 | -0.55 | -0.63 | -0.44 | -0.50 | -0.50 | -0.49 | 0.36 | -0.07 | 0.46 | 0.53 | 1.00 | 0.40 | -0.32 | 0.38 |
| R@1 | -0.67 | -0.46 | -0.36 | -0.66 | -0.57 | -0.57 | -0.78 | 0.29 | -0.35 | 0.74 | 0.62 | 0.40 | 1.00 | -0.57 | -0.07 |
| R@10 | 0.36 | 0.25 | 0.12 | 0.52 | 0.28 | 0.28 | 0.42 | -0.22 | 0.75 | -0.45 | -0.14 | -0.32 | -0.57 | 1.00 | 0.37 |
| R@100 | -0.28 | -0.55 | -0.72 | 0.39 | 0.24 | 0.23 | -0.28 | 0.51 | 0.64 | -0.23 | 0.02 | 0.38 | -0.07 | 0.37 | 1.00 |

Table 19: Correlation between metrics in AMLB Regression dataset.

| | N@1 | N@10 | N@100 | M@1 | M@10 | M@100 | S@1 | S@10 | S@100 | T@1 | T@10 | T@100 | R@1 | R@10 | R@100 |
|---|---|---|---|---|---|---|---|---|---|---|---|---|---|---|---|
| N@1 | 1.00 | 0.93 | 0.87 | 0.88 | 0.84 | 0.88 | 0.87 | 0.67 | 0.48 | -0.86 | -0.74 | -0.68 | -0.81 | -0.71 | -0.77 |
| N@10 | 0.93 | 1.00 | 0.92 | 0.81 | 0.87 | 0.91 | 0.82 | 0.61 | 0.50 | -0.79 | -0.84 | -0.76 | -0.86 | -0.73 | -0.73 |
| N@100 | 0.87 | 0.92 | 1.00 | 0.88 | 0.95 | 0.95 | 0.87 | 0.77 | 0.66 | -0.87 | -0.90 | -0.80 | -0.81 | -0.85 | -0.84 |
| M@1 | 0.88 | 0.81 | 0.88 | 1.00 | 0.95 | 0.95 | 0.96 | 0.85 | 0.68 | -1.00 | -0.82 | -0.72 | -0.75 | -0.81 | -0.83 |
| M@10 | 0.84 | 0.87 | 0.95 | 0.95 | 1.00 | 0.99 | 0.92 | 0.81 | 0.70 | -0.94 | -0.91 | -0.80 | -0.76 | -0.83 | -0.83 |
| M@100 | 0.88 | 0.91 | 0.95 | 0.95 | 0.99 | 1.00 | 0.92 | 0.78 | 0.66 | -0.94 | -0.90 | -0.81 | -0.80 | -0.82 | -0.83 |
| S@1 | 0.87 | 0.82 | 0.87 | 0.96 | 0.92 | 0.92 | 1.00 | 0.81 | 0.71 | -0.96 | -0.84 | -0.68 | -0.71 | -0.77 | -0.79 |
| S@10 | 0.67 | 0.61 | 0.77 | 0.85 | 0.81 | 0.78 | 0.81 | 1.00 | 0.82 | -0.87 | -0.72 | -0.57 | -0.54 | -0.81 | -0.83 |
| S@100 | 0.48 | 0.50 | 0.66 | 0.68 | 0.70 | 0.66 | 0.71 | 0.82 | 1.00 | -0.71 | -0.81 | -0.60 | -0.35 | -0.62 | -0.72 |
| T@1 | -0.86 | -0.79 | -0.87 | -1.00 | -0.94 | -0.94 | -0.96 | -0.87 | -0.71 | 1.00 | 0.83 | 0.72 | 0.73 | 0.82 | 0.83 |
| T@10 | -0.74 | -0.84 | -0.90 | -0.82 | -0.91 | -0.90 | -0.84 | -0.72 | -0.81 | 0.83 | 1.00 | 0.88 | 0.74 | 0.80 | 0.81 |
| T@100 | -0.68 | -0.76 | -0.80 | -0.72 | -0.80 | -0.81 | -0.68 | -0.57 | -0.60 | 0.72 | 0.88 | 1.00 | 0.84 | 0.83 | 0.82 |
| R@1 | -0.81 | -0.86 | -0.81 | -0.75 | -0.76 | -0.80 | -0.71 | -0.54 | -0.35 | 0.73 | 0.74 | 0.84 | 1.00 | 0.87 | 0.72 |
| R@10 | -0.71 | -0.73 | -0.85 | -0.81 | -0.83 | -0.82 | -0.77 | -0.81 | -0.62 | 0.82 | 0.80 | 0.83 | 0.87 | 1.00 | 0.86 |
| R@100 | -0.77 | -0.73 | -0.84 | -0.83 | -0.83 | -0.83 | -0.79 | -0.83 | -0.72 | 0.83 | 0.81 | 0.82 | 0.72 | 0.86 | 1.00 |

## B.5 BONFERRONI CORRECTION

Table 20 describes the experimental results using the Bonferroni correction for each metric, considering the number of independent tests as the product of the number of seeds, the number of tasks, and the number of models evaluated for the metric. The column "p-value" contains the p-value obtained after performing the Wilcoxon signed-rank test between the score-based and rank-based versions. The column "$\alpha_{\text{Bonferroni}}$" contains the corrected value (i.e., $\alpha/\#$ independent tests, with $\alpha = 0.05$). Finally, the column "$<$" indicates whether the p-value is less than $\alpha_{\text{Bonferroni}}$, meaning that the null hypothesis—that the metric value obtained by the rank-based version is less than or equal to that of the score-based version—is rejected.

Table 20: Results of the statistical significance tests for each metric after Bonferroni correction

| | #independent tests ($seeds * tasks * models$) | p-value | $\alpha_{\text{Bonferroni}}$ | diff | $< 0$ |
|---|---|---|---|---|---|
| NDCG@1 | 17760 | 0.00e+0 | 2.82e-6 | -2.82e-6 | yes |
| NDCG@5 | 8400 | 7.56e-239 | 5.95e-6 | -5.95e-6 | yes |
| NDCG@10 | 17760 | 0.00e+0 | 2.82e-6 | -2.82e-6 | yes |
| NDCG@100 | 9360 | 0.00e+0 | 5.34e-6 | -5.34e-6 | yes |
| MRR@1 | 17760 | 5.26e-30 | 2.82e-6 | -2.82e-6 | yes |
| MRR@5 | 8400 | 1.21e-43 | 5.95e-6 | -5.95e-6 | yes |
| MRR@10 | 17760 | 6.84e-117 | 2.82e-6 | -2.82e-6 | yes |
| MRR@100 | 9360 | 2.72e-62 | 5.34e-6 | -5.34e-6 | yes |
| SCORE@1 | 17760 | 1.83e-195 | 2.82e-6 | -2.82e-6 | yes |
| SCORE@5 | 8400 | 3.22e-100 | 5.95e-6 | -5.95e-6 | yes |
| SCORE@10 | 17760 | 6.10e-11 | 2.82e-6 | -2.82e-6 | yes |
| SCORE@100 | 9360 | 1.10e-13 | 5.34e-6 | -5.34e-6 | yes |
| TTB@1 | 9360 | 2.47e-28 | 5.34e-6 | -5.34e-6 | yes |
| TTB@10 | 9360 | 2.54e-23 | 5.34e-6 | -5.34e-6 | yes |
| TTB@100 | 9360 | 2.35e-78 | 5.34e-6 | -5.34e-6 | yes |
| AVG_RANK@1 | 17760 | 2.34e-259 | 2.82e-6 | -2.82e-6 | yes |
| AVG_RANK@5 | 8400 | 3.13e-100 | 5.95e-6 | -5.95e-6 | yes |
| AVG_RANK@10 | 17760 | 3.14e-59 | 2.82e-6 | -2.82e-6 | yes |
| AVG_RANK@100 | 9360 | 7.23e-25 | 5.34e-6 | -5.34e-6 | yes |

# C    PIPELINE SEARCH SPACE

We describe the search space for machine learning pipelines as a grammar, with non-terminal symbols in ALL CAPS and terminal symbols in CamelCase. This encompasses the various components that can be assembled to preprocess data, select features, and construct models for classification or regression tasks. The structure of the pipeline is modular, allowing for different preprocessing techniques for categorical and numerical data, various feature selection methods, and a wide range of classifiers and regressors. A simplified grammar is described below.

$$\text{PIPELINE} := \text{DATA\_PREPROCESS} \,\&\, \text{FEATURE\_SELECTOR} \,\&\, (\text{CLASSIFIER} \mid \text{REGRESSOR})$$

$$\text{DATA\_PREPROCESS} := \text{CATEGORICAL} \mid \text{NUMERICAL} \mid (\text{CATEGORICAL} \,\&\, \text{NUMERICAL})$$

$$\text{CATEGORICAL} := (\text{CategoricalImputation} \,\&\, \text{ENCODING}) \mid \text{ENCODING}$$

$$\text{ENCODING} := \text{NoEncoding} \mid \text{OneHotEncoder}$$

$$\text{NUMERICAL} := (\text{NumericalImputation} \,\&\, \text{SCALING}) \mid \text{SCALING}$$

$$\text{SCALING} := \text{NoRescaling} \mid \text{StandardScaler} \mid \text{MinMaxScaler} \mid$$
$$\text{Normalizer} \mid \text{QuantileTransformer} \mid \text{RobustScaler}$$

$$\text{FEATURE\_SELECTOR} := \text{NoPreprocessing} \mid \text{Densifier} \mid \text{ExtraTreesPreprocessor} \mid$$
$$\text{FastICA} \mid \text{FeatureAgglomeration} \mid \text{KernelPCA} \mid$$
$$\text{RandomKitchenSinks} \mid \text{LibLinear} \mid \text{Nystroem} \mid$$
$$\text{PCA} \mid \text{PolynomialFeatures} \mid \text{RandomTreesEmbedding} \mid$$
$$\text{SelectPercentile} \mid \text{SelectClassificationRates} \mid \text{TruncatedSVD}$$

$$\text{CLASSIFIER} := \text{AdaboostClassifier} \mid \text{BernoulliNBClassifier} \mid$$
$$\text{DecisionTreeClassifier} \mid \text{ExtraTreesClassifier} \mid$$
$$\text{GaussianNBClassifier} \mid \text{GradientBoostingClassifier} \mid$$
$$\text{KNearestNeighborsClassifier} \mid \text{LDAClassifier} \mid$$
$$\text{LibLinear\_SVCClassifier} \mid \text{LibSVM\_SVCClassifier} \mid$$
$$\text{MultinomialNBClassifier} \mid \text{PassiveAggressiveClassifier} \mid$$
$$\text{QDAClassifier} \mid \text{RandomForestClassifier} \mid \text{SGDClassifier} \mid \text{MLPClassifier}$$

$$\text{REGRESSOR} := \text{AdaboostRegressor} \mid \text{ARDRegressor} \mid$$
$$\text{DecisionTreeRegressor} \mid \text{ExtraTreesRegressor} \mid$$
$$\text{GaussianProcessRegressor} \mid \text{GradientBoostingRegressor} \mid$$
$$\text{KNearestNeighborsRegressor} \mid \text{LibSVM\_SVRegressor} \mid$$
$$\text{MLPRegressor} \mid \text{RandomForestRegressor} \mid \text{SGDRegressor}$$

