# OpenReview forum: "Learning to Rank for AutoML: enhancing pipeline selection with ranking information"
_ICLR.cc/2025/Conference — ICLR 2025 Conference Withdrawn Submission_

### Official Review · Reviewer_p4ta · 2024-10-29

**Soundness:** 2
**Presentation:** 3
**Contribution:** 1
**Rating:** 3
**Confidence:** 3

**Summary:**

The paper introduces a learning-to-rank (LTR) framework to improve pipeline selection in automated machine learning (AutoML). Traditional AutoML methods predict performance metrics to select optimal pipelines, which can be limiting and resource-intensive. Instead, this framework uses ranking-based selection, focusing on ranking pipelines by performance potential rather than estimating specific metrics. By aligning selection with ranking metrics such as NDCG and MRR, this method creates a more robust, metric-agnostic approach.

The paper evaluates LTR with sequential optimization techniques like Bayesian optimization and Monte Carlo tree search on OpenML datasets, finding significant improvements over the score-based methods. Key contributions include:
* Introducing LTR approach for AutoML pipeline selection.
* An empirical comparison demonstrating the advantages of ranking-based over score-based methods.
* Insights into the correlation between ranking metrics and traditional AutoML objectives.

**Strengths:**

1. Paper is well written, pleasing-to-read.
2. Comprehensive experiments, with details available in the appendix.

**Weaknesses:**

1. I personally enjoyed the paper and the work of replacing autoML task with learning to rank. However, my main worry is that the impact of the work could be limited. I might be biased. But my feeling is that attentions have been attacked to LLM and related works. The paper might be better justified for its impact if authors can show some evidence like introducing LTR improves AutoML for LLM training or prompting, etc. This may well beyond the scope of the work.
2. MCTS results missing on OpenML-Weka in Figure 1. Didn’t find any explanation for the missing results there either.
3. Minor issue: “green > 0, dark green > 10” in Line 347 and “yellow < 0, orange < 10”  in Line 348 should be better with percent signs like “green > 0, dark green > 10%” and “yellow < 0, orange < 10%”.

**Questions:**

1. Why rank-based approach show consistently (slightly) worse performance on D2 AMLB classification task in Table 2? Any intuition? Can rank-based approach be useful for specific tasks?

---

> ### Author Response · Authors · 2024-11-26
> **Response to Reviewer p4ta**
>
> Thank you for your thoughtful review and positive comments regarding the readability and comprehensive nature of our paper. We appreciate your insights.
>
> We understand your concern regarding the impact of LTR within the current landscape, especially with the recent focus on LLMs and related advancements. However, as you noted, this direction—applying LTR for pipeline selection in AutoML—remains relevant to improving AutoML frameworks and could potentially extend to tasks related to LLM optimization in the future. While this area was outside our current scope, we agree that exploring LTR's applications in LLM training or prompting tasks is an exciting avenue for future work.
>
> The absence of MCTS results for OpenML-Weka in Figure 1 occurred because MCTS requires knowledge of pipeline components to structure the exploration as a decision tree. For this dataset, we lacked that information, making it impossible to run MCTS. We will clarify this in the revised paper to ensure transparency and reduce any potential confusion.
>
> We appreciate your suggestion to add percent signs for clarity on color-coded values in Table 2. We will incorporate these changes in the revised draft to improve readability.
>
> Regarding the performance on D2 (AMLB classification task), we acknowledge that rank-based methods sometimes show slightly lower performance on specific tasks. However, in most cases, the difference is not statistically significant. This occurs because performance in a few isolated tasks may skew the average, as outliers shift the metric. While it could be interesting to investigate the specific task causing this, in AutoML, we generally avoid focusing too closely on individual tasks to prevent dataset-specific overfitting and biases. Instead, we treat the task suite as a closed set.
>
> Thank you again for your feedback, and we hope these clarifications address your concerns. For additional comments regarding the novelty of our work, please refer to the general comment “To All Reviewers” addressed to all reviewers.

---

> ### Comment · Reviewer_p4ta · 2024-11-27
>
> Thank you for your response. Great to see some suggestions have been considered.
>
> However, nothing significant has changed. I would stick to my current scoring as well.

---

> > ### Author Response · Authors · 2024-11-27
> >
> > Thank you for your quick response.
> >
> > We believe we have addressed all major and minor concerns raised by reviewers, some of which originated in some misunderstandings around automl and its challenges. As such, we tried to provide such context and to point out why the improvements shown by our approach are non-trivial and impactful.
> >
> > We don’t see what we would have changed (significantly) to meet the reviewer’s expectations of our work since, as we understand, the only major observation is related to the use of LLMs that -as the reviewers acknowledged- is beyond the scope of our work.
> > We kindly ask the reviewer to reconsider his/her evaluation in light of the paper's scope and shown contributions.

---

### Official Review · Reviewer_Vi55 · 2024-11-01

**Soundness:** 3
**Presentation:** 3
**Contribution:** 2
**Rating:** 5
**Confidence:** 2

**Summary:**

This paper proposes an approach for AutoML which is based on learning the rank/position of candidate configurations rather than their absolute performance score. Multiple regression models are evaluated for learning to map a configuration to a position/rank. Experiments on several AutoML benchmarks show improved performance compared to score-based models.

* I am not an expert on AutoML, not sure why this paper was assigned to me. I can only provide a low-confidence review.

**Strengths:**

* The empirical results seem appealing in terms of achieving better performance compared to a score-based approach.
* The paper is clearly written.

**Weaknesses:**

* The proposed approach is using a regression model with ranks instead of scores, which is different from a ranking algorithm which outputs a permutation. The authors state this clearly in the paper, however they do sometimes call this a learning-to-rank problem (e.g., in the title and in line 512), which often refers to ranking losses (pairwise/listwise) and not regression using ranks. It may be interesting to see how a ranking loss based approach would compare to the regression based one.
* The proposed approach covers only search spaces with a predefined set of configurations that is enumerable. Large search spaces are not handled.
* The use of positions instead of performance scores seems somewhat incremental (replacing scores by ranks).

**Questions:**

* Table 2: is there an explanation of the missing entries for D1 in the table? Maybe I missed it.
* In Related Work, there is some recent work from neural architecture search, a related problem in AutoML, using ranking instead of scoring for training predictors. Some examples are:
  * Renas: Relativistic evaluation of neural architecture search, Xu et al. (2021)
  * DCLP: Neural Architecture Predictor with Curriculum Contrastive Learning, Zheng et al. (2024)
  * FlowerFormer: Empowering Neural Architecture Encoding using a Flow-aware Graph Transformer, Hwang et al. (2024)
Can you please comment on how this relates to your work?

Minor/typos:
* Line 135: perhaps give MSE (s_ij - f_ij)^2 as an example loss?
* Line 255: “This set of OpenML tasks includes 104 tasks, split *evenly* between 71 classification and 33 regression problems” — what do you mean by “evenly”?
* Table 2: Consider adding a legend for colors in addition to or instead of the caption description. Also, consider sorting first by dataset and then by model, it seems like the similarity within a dataset is higher than within a model. This would also be consistent with the arrangement figures 1 and 2.

---

> ### Author Response · Authors · 2024-11-26
> **Response to Reviewer Vi55**
>
> Thank you for your review and for the positive comments regarding our empirical results and clarity of presentation. We appreciate your feedback, and even with limited familiarity in AutoML, we thank again for your time and effort in reviewing our work.
>
> We appreciate your observation about the distinction between regression-based ranking and traditional ranking algorithms. We agree that the title may suggest a broader focus, and we might reformulate it if needed. Pairwise and listwise approaches were outside the scope of our study due to their computational demands, allowing us to maintain focus on a regression-based approach that aligns well with practical AutoML workflows. We aimed to keep the initial exploration simple, demonstrating that even a basic rank transformation in regression offers substantial improvements. This foundational work could indeed open the door for future studies.
>
> Your point regarding the limitation to enumerated search spaces is well taken. Our decision to begin with predefined spaces aligns with a trend in AutoML applications that tend to work with predefined sets of pipelines and default hyperparameters. Also, this setup is rather common in the practice of AutoML [A]. Exploring broader, dynamically expanding search spaces is indeed an interesting future direction, and we see our work as an initial step toward applying ranking techniques in more general settings.
>
> In terms of novelty, we would like to clarify that our use of ranks rather than scores is not merely an incremental substitution. This shift allows for metric-agnostic optimization that circumvents the need to model each metric individually, reducing computational and developmental overheads while achieving better alignment with ranking-focused objectives, which, as we demonstrated, correlate directly with the ultimate AutoML goal of finding the best configuration as quickly as possible.
>
> Regarding Table 2, the missing entries for D1 have different reasons. On one hand, the dataset D1 lacks temporal information, making it impossible to calculate TTB. On the other hand, it also lacks information about the components that form the pipelines, which means that MCTS cannot be executed on this dataset, as MCTS requires knowledge of pipeline components to structure exploration as a decision tree.
>
> We appreciate the suggestion to reference recent NAS work. While NAS and pipeline selection remain distinct areas within AutoML, we agree that referencing this related work could help contextualize our contributions and broaden the impact of our discussion. We will consider how these NAS ranking techniques compare with our approach and discuss this in future work.
>
> Minor Corrections and Formatting Suggestions:
> - We incorporate MSE as an example loss function for clarity.
> - We replace the wording "evenly split" by just “split” in Line 255 .
> - For Table 2, we appreciate your suggestions regarding color legend and sorting by dataset. We will revise the formatting accordingly to enhance readability and alignment with Figures 1 and 2.
>
> We hope our clarifications have addressed your concerns and provided a clearer perspective on the contributions of our work. For additional comments regarding the novelty of our work, please refer to the general comment “To all reviewers” addressed to all reviewers.
>
> [A] Erickson, N., Mueller, J., Shirkov, A., Zhang, H., Larroy, P., Li, M., & Smola, A. (2020). Autogluon-tabular: Robust and accurate automl for structured data.

---

> > ### Comment · Reviewer_Vi55 · 2024-12-03
> >
> > Thank you for the clarifications.

---

### Official Review · Reviewer_7X7T · 2024-11-04

**Soundness:** 3
**Presentation:** 2
**Contribution:** 2
**Rating:** 5
**Confidence:** 4

**Summary:**

Overall the paper proposes an improvement in Automation of ML pipeline by predicting rank of an approach in the pool given config and data instead of predicting a metric score or formulating as a true ranking problem. They show promising results on different datasets and metrics. The experiment section is fairly elaborate.

**Strengths:**

Fairly elaborate experiment section.
Simple change to existing approach.

**Weaknesses:**

Writing can be improved significantly. It is a bit hard to read and understand the contributions as they are mixed with previous work and formulation.
Novelty is low.

**Questions:**

It is said that the approach is metric agnostic, you say that because the approach only cares about rank and not the metric value right? even though thats the case the rank order is determined by the metric right?? Would is still be metric agnostic? or is it metric-value agnostic?

In experiments given you are predicting rank order only would it make sense to try some classifier also instead of just regressors??

Would it also make sense to compare your approach to a model trained with some ranking loss function instead only comparing with score-based approach??

In BO approach used, you mention its pretrained, can you clarify how?

---

> ### Author Response · Authors · 2024-11-26
> **Response to Reviewer 7X7T**
>
> First, we would like to thank the reviewer for their review and for highlighting our experiment section.
>
> Regarding our contribution, our paper proposes a distinct shift from traditional score-prediction methods to a rank-based approach in the context of AutoML pipeline selection, representing a novel paradigm in this field. This shift simplifies comparisons across various AutoML tasks, as shown in our results using ranking metrics (NDCG, MRR), which align more closely with real-world AutoML objectives. We strongly believe this is not an incremental change but rather a fundamental shift in how we pre-select pipelines within AutoML which allows us to address problems of a different nature (classification, regression, among others) within the same conceptual framework. Furthermore, the discovery of the correlation between ranking metrics (NDCG and MRR) and AutoML system metrics (Score and TTB) is significant, as it enables optimization by directly addressing the ranking problem. This approach translates to savings in cost, energy, and the need for specialized models for individual metrics. Finally, the introduction of BO-Rank and MCTS-Rank are also important contributions since, to the best of our knowledge, these variants do not exist in the AutoML literature. Transforming score regression into a ranking problem using position could inspire new versions of approaches applicable to other domains, thereby impacting other areas.
>
> Regarding whether the approach is metric-agnostic or metric-value-agnostic, the metric value is important because it determines the ranking position, so the model cannot be metric-value agnostic. Instead, this approach allows us to work similarly whether the metric is Accuracy or NMSE, making it metric-agnostic rather than metric-value agnostic.
>
> We excluded approaches that use classification loss or ranking loss from the scope of this work to avoid excessive complications in the experimental setup, since they  significantly increase computational costs. We opted to use regressors, as they are the most common choice in AutoML system design. By simply transforming the target to its position, we demonstrate that effective results can be achieved without the need for more computationally complex models.
>
> In the BO approach, when we mention "pre-trained," we mean that we initially trained a score-based linear regressor and a ranking-based one using the training tasks. Then, as we obtained more data, we continued to retrain them. Details on this process are provided in the code we attached, but we agree that clarifying this aspect in the paper is necessary, as the regressor component may be unclear.
>
> Finally, we are a bit surprised that the reviewer found our writing unsatisfactory, as other reviewers highlighted it as a strength. We apologize if our contributions were unclear and kindly ask the reviewer for suggestions, as they were explicitly listed as bullet points at the end of the introduction.
>
> We hope our clarifications have addressed your concerns and provided a clearer perspective on the contributions of our work. For additional comments regarding the novelty of our work, please refer to the general comment “To All Reviewers” addressed to all reviewers.

---

> ### Comment · Reviewer_7X7T · 2024-11-27
> **Thanks for the response.**
>
> Thank you for the response. The comments addresses many of my concerns. Below is my response and the concerns still remain.
> * I would agree that is a simple yet neat idea and this shift improves on some of the existing approach. But the lack of comparison against a ranking-based approach dimish the strength of the approach, even if the previous ranking-based approaches are slow etc, having it to compare will only strengthen the approach further in my opinion making it easy to fully appreciate the beauty of the approach. As i mentioned earlier, the simple yet effective approach is a strength but needs a bit more IMO.
> * About the writing, my opinion was that the way it is written Previous approaches + problem set up and your contributions are mixed together in main section. Even if some people may have found it easy to read i personally felt that it makes it hard to find where the details of your contribution/novelty begins and ends. It may be worth organizing the main section to flow in a way that it makes it easy to make that distinction.
>
> Thanks again for the response. Based on the response i will update my scores. Thanks

---

### Note · Authors · 2025-06-04

I have read and agree with the venue's withdrawal policy on behalf of myself and my co-authors.

---

### Meta-Review · Area_Chair_ddcS · 2024-12-21

**Metareview:**

This paper proposes a learning-to-rank framework for AutoML pipeline selection which changes the traditional score prediction approach to ranking-based optimization. By focusing on pipeline ranking, the approach offers a more direct, efficient metric-agnostic solution. The comprehensive experiments across diverse datasets demonstrate improvements in ranking metrics such as NDCG and MRR. They also explored the integration of LTR with Bayesian optimization and Monte Carlo tree search, which further enhance the method’s performance.
The strengths of this paper is its clear motivation and detailed experimental evaluation. However, the novelty is somewhat limited, as the primary contribution involves a shift to rank-based optimization without sufficient comparison to existing ranking-based approaches. Additionally, the method is restricted to predefined search spaces, limiting its applicability to larger or dynamic configurations. These issues undermine the impact of the present paper.

**Additional Comments On Reviewer Discussion:**

During the discussion period, the authors clarified the unique features of their approach and provided additional context for their experimental setup. They helped address some concerns and improved the reviewers’ understanding of the proposed method. However, the key concerns about the lack of novelty and insufficient comparisons to existing ranking methods were not resolved. These unresolved issues weakened the overall evaluation of the paper.

---

### Decision · Program_Chairs · 2025-01-22

Reject